# Oncofetal gene *SALL4* reactivation by hepatitis B virus counteracts miR-200c in PD-L1-induced T cell exhaustion

Cheng Sun[1,2], Peixiang Lan[3], Qiuju Han[3], Mei Huang[4], Zhihong Zhang[5], Geliang Xu[4], Jiaxi Song[1], Jinyu Wang[1], Haiming Wei[1], Jian Zhang[3], Rui Sun[1], Cai Zhang[3] & Zhigang Tian[1,2]

A chronic viral or tumor microenvironment can push T cells to exhaustion by promoting coinhibitory ligand expression. However, how host factors control coinhibitory ligand expression and whether viral infection breaks this control during tumor progress is unknown. Here we show a close negative correlation between *SALL4* or PD-L1 and miR-200c in tumors from 98 patients with HBV-related hepatocellular carcinoma. SALL4 or PD-L1 expression correlates negatively with miR-200c expression, and patients with lower levels of SALL4 or PD-L1 and higher miR-200c survive longer. Moreover, over-expression of miR-200c antagonizes HBV-mediated PD-L1 expression by targeting 3′-UTR of *CD274* (encoding PD-L1) directly, and reverses antiviral CD8$^+$ T cell exhaustion. MiR-200c transcription is inhibited by oncofetal protein SALL4, which is re-expressed through HBV-induced STAT3 activation in adulthood. We propose that an HBV-pSTAT3-SALL4-miR-200c axis regulates PD-L1. Therapeutic strategies to influence this axis might reverse virus-induced immune exhaustion.

[1] The CAS Key Laboratory of Innate Immunity and Chronic Disease and Institute of Immunology, School of Life Science and Medical Center, University of Science and Technology of China, Hefei 230027 Anhui, China. [2] Collaborative Innovation Center for Diagnosis and Treatment of Infectious Diseases, State Key Laboratory for Diagnosis and Treatment of Infectious Diseases, First Affiliated Hospital, College of Medicine, Zhejiang University, Hangzhou 310003 Zhejiang, China. [3] Institute of Immunopharmacology and Immunotherapy, School of Pharmaceutical Sciences, Shandong University, Jinan 250012 Shandong, China. [4] Anhui Province Key Laboratory of Hepatopancreatobiliary Surgery, Anhui Provincial Hospital Affiliated with Anhui Medical University, Hefei 230001 Anhui, China. [5] Britton Chance Center for Biomedical Photonics, Wuhan National Laboratory for Optoelectronics, Huazhong University of Science and Technology, Wuhan 430074 Hubei, China. These authors contributed equally: Cheng Sun, Peixiang Lan, Qiuju Han. Correspondence and requests for materials should be addressed to C.Z. (email: caizhangsd@sdu.edu.cn) or to Z.T. (email: tzg@ustc.edu.cn)

Chronic viral infection and tumor microenvironments can push infiltrating virus-specific or tumor-specific T cells to exhaustion such that proliferative capacity and effector functions of these cells are severely impaired, rendering the immune response unable to eliminate the virus or to reject the tumors. Although T cell coinhibitory receptors were first identified via their prevention of autoimmunity in mice, these receptors are now thought to be critical regulators of T cell exhaustion in the context of chronic viral infections and tumors. A breakthrough in cancer immunotherapy for antagonizing T cell exhaustion is to reactivate immune responses by blocking the inhibitory signals (e.g., programmed cell death-ligand 1, PD-L1) with antibodies. The success of blockade has demonstrated that coinhibitory signaling that restrains the activation and function of effector lymphocytes is a checkpoint for immunotherapeutic reversal of immune cell exhaustion. However, the interactions between chronic viral infection, tumorigenesis, and coinhibitory ligand expression is unclear.

The activation and maintenance of CD8[+] T cell clones is critical in clearance of cancer and viral infections, such as hepatitis B virus (HBV), hepatitis C virus (HCV), and human cytomegalovirus[1,2]. Hepatic viral infections are the major factors in promoting the development and progression of hepatocellular carcinoma (HCC). The majority of HCC cases are reported to be the result of persistent HBV or HCV infection. CD8[+] T cell activation is regulated not only by recognizing epitopes presented on the surfaces of infected hepatocytes, but also by a balance between positive and negative signals mediated by the interaction of coinhibitory and costimulatory molecules on the T cell surface with ligands on antigen presenting cells (APCs) including hepatocytes, which mostly determines the outcome of T cell activation and subsequent effector functions[2–4]. PD-1 is expressed on activated T and B cells as an inhibitory receptor, mediating negative signals for T cell activation[5]. CD8[+] T cells from PD-1-deficient mice have increased proliferative capacity and enhanced antiviral responses to adenovirus infection[6]. PD-L1, the ligand of PD-1, is expressed on numerous cell types, such as dendritic cells, macrophages, hepatocytes, and tumor cells[5]. In both chronic HBV infections (CHB) and HBV-related HCC patients, antiviral T cell responses are markedly impaired and T cells are prone to apoptosis, characterized by low secretion of IFN-γ and TNF and a high expression of PD-1[7,8]. Clinical data also show that PD-1/PD-L1 expression is positively associated with tumor size, blood vessel invasion, and tumor stage classification in patients with HCC[8,9]. Therefore, induction of PD-L1 expression by hepatocytes and subsequent high PD-1 expression by CD8[+] T cells is considered to have a critical function in CD8[+] T cell exhaustion. To date, how HBV infection induces PD-L1 expression, whether host factors control HBV-induced PD-L1 expression, and potential interplay mechanisms, are unclear.

MicroRNAs (miRNA) regulate target genes post-transcriptionally by directing the degradation and/or repression of the translation of mRNA, leading to a reduction in protein levels[10]. Evidence indicates that miRNAs regulate the host antiviral immune response and miRNAs are considered to be potential biomarkers for the prognosis of HBV-related HCC. For example, miR-96 and miR-372/373 were elevated in HBV-associated HCC, and contribute to the progression of HBV[+] HCC[10]. In another study, over-expression of miR-155 was shown to enhance the antiviral immune responses to HBV[11]. The miR-141 and miR-200 family groups were down-regulated in HCC with bile duct tumor thrombus and act as independent predictors for disease-free survival[12]. For the regulation of PD-L1, miR-513 has been shown to regulate PD-L1 expression in response to IFN-γ or Cryptosporidium parvum infection[13]. However, whether miRNAs are involved in the regulation of PD-L1 expression and

whether HBV counteracts intrinsically with miRNAs, and how the interplay affects anti-HBV immunity needs to be investigated.

Sal-like protein 4 (Sall4) is a zinc finger transcription factor that regulates the pluripotency and self-renewal of embryonic stem cells[14,15]. SALL4 is expressed in human fetal liver, but not in healthy adult liver; however, SALL4 is re-expressed in a number of human cancers, particularly HCC. Moreover, high expression of SALL4 is associated with aggressive HCC and poor prognosis in clinical investigations[16–18]. However, whether HBV has a function in reactivating SALL4 expression in adult liver, and whether SALL4 and miRNA(s) interact to regulate PD-L1 expression is not clear.

In this study, we show a close negative correlation between SALL4 or PD-L1 and miR-200c, with corresponding survival after analysis of tumors from 98 patients with HCC. We also show that miR-200c controls PD-L1 expression by directly targeting the 3′-UTR. Via over-expression of miR-200c or injection with miR-200c mimics, we further confirm that miR-200c can directly inhibit PD-L1 expression by hepatocytes and restore the dysfunction of CD8[+] T cells in vivo. Interestingly, HBV might antagonize miR-200c function by reactivating transcriptional repressor SALL4 through STAT3 pathway, leading to an increase in PD-L1 expression. We propose that a HBV-pSTAT3-SALL4-miR-200c axis exists in the regulation of PD-L1 expression during HBV infection and hepatoma progression, and therapeutic strategies to interrupt this axis might reverse virus-induced immune exhaustion.

## Results

**SALL4 and PD-L1 negatively correlate with miR-200c in HCC**. The scientific community recognizes that PD-L1 plays a role in multiple malignancies, and miR-200c has been reported as one of the miRNA indicators for HCC in a microarray analysis[19], and also SALL4 is found to be highly expressed in HCC with poor prognosis[16,17]. Using a miRNA analysis tool (www.microrna.org), we found that miR-200c directly targets the 3′-UTR of PD-L1, and on the other hand, transcription factor SALL family regulates miR-200c expression as for three putative binding sites within the promoter region of miR-200c (Fig. 1a). Utilizing the tumor tissues from 98 patients with HCC (Supplementary Table 1) to show the expression of PD-L1 and SALL4 in the peritumor and center tumor regions by immunostaining (Fig. 1b), we found that the expression of both SALL4 and PD-L1 was significantly increased in the center tumor regions (P < 0.0001, two-tailed unpaired Student's t-test) (Fig. 1c, d). By employing in situ hybridization to show the expression of miR-200c, two representative sections were exhibited (Fig. 1b), and a statistically significant higher density of miR-200c in tumors than in peritumor regions (P < 0.0001, two-tailed unpaired Student's t-test) was also observed (Fig. 1e). Notably, although SALL4, PD-L1, and miR-200c significantly increased, these three molecules experienced extreme disparity, with a much higher SALL4 or PD-L1 expression and a relatively lower miR-200c in the center tumor regions when compared with those in peritumor regions, while SALL4 and PD-L1 were paralleled in expression (Fig. 1f), raising the possible close relations among SALL4, miR-200c, and PD-L1 during HCC progression.

We analyzed the relationship among SALL4, miR-200c, and PD-L1 expression in the center tumor regions from each HCC patient. We found that lower expressions of miR-200c clearly accompanied with a higher density of either SALL4 or PD-L1, while a higher expression of miR-200c maintained a lower density of either SALL4 or PD-L1 in three representative patients (Fig. 2a), leading to highly significant two pairs of negative correlations between SALL4 and miR-200c (P = 0.002,

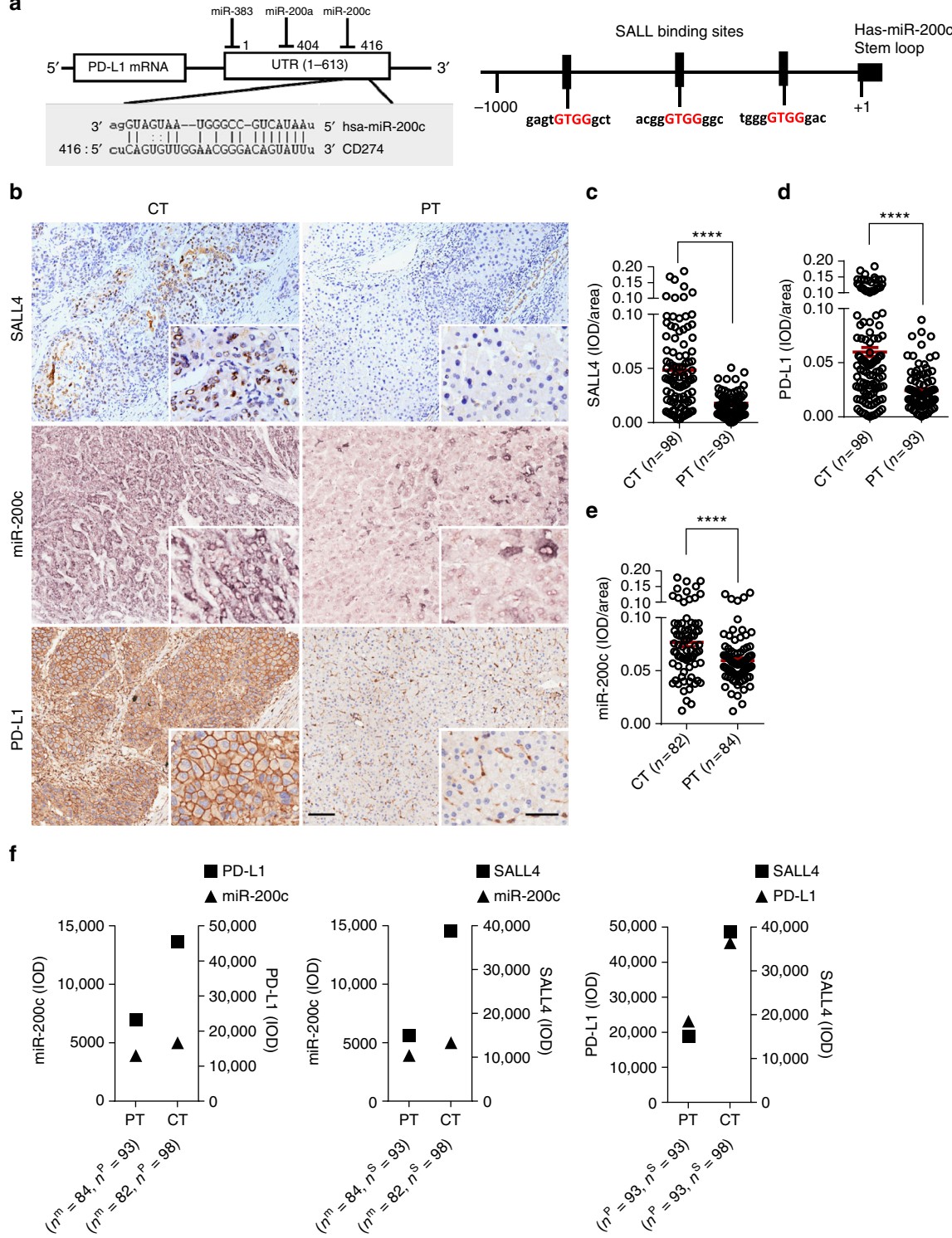

**Fig. 1** Expression of SALL4, miR-200c, and PD-L1 in the center tumor regions of HCC. **a** Predicted microRNA targeting *PD-L1* 3′-UTR includes miR-200a, miR-200c, and miR-383 (left); the upstream promoter region of the human miR-200c showing putative SALL4-binding sites (right). **b** Immunostaining of SALL4 and PD-L1 proteins (brown color) and in situ hybridization for hsa-miR-200c (blue purple color) in the center of the tumor (CT) and peritumor (PT) regions of HCC. Original magnifications: ×10, ×20; bar = 100 μm, 50 μm, respectively. **c**–**e** Increased SALL4 (**c**, $P < 0.0001$), PD-L1 (**d**, $P < 0.0001$) and miR-200c expression (**e**, $P < 0.0001$) in CT regions. Cumulative data calculated by two-tailed unpaired Student's $t$-test. The results are expressed as the mean ± SEM. ****$P < 0.0001$. **f** Disparity analysis between miR-200c and PD-L1 (left), miR-200c and SALL4 (middle), PD-L1 and SALL4 (right) in CT and PT regions. $n^P$, number of patients with PD-L1; $n^m$, number of patients with miR-200c; $n^S$, number of patients with SALL4

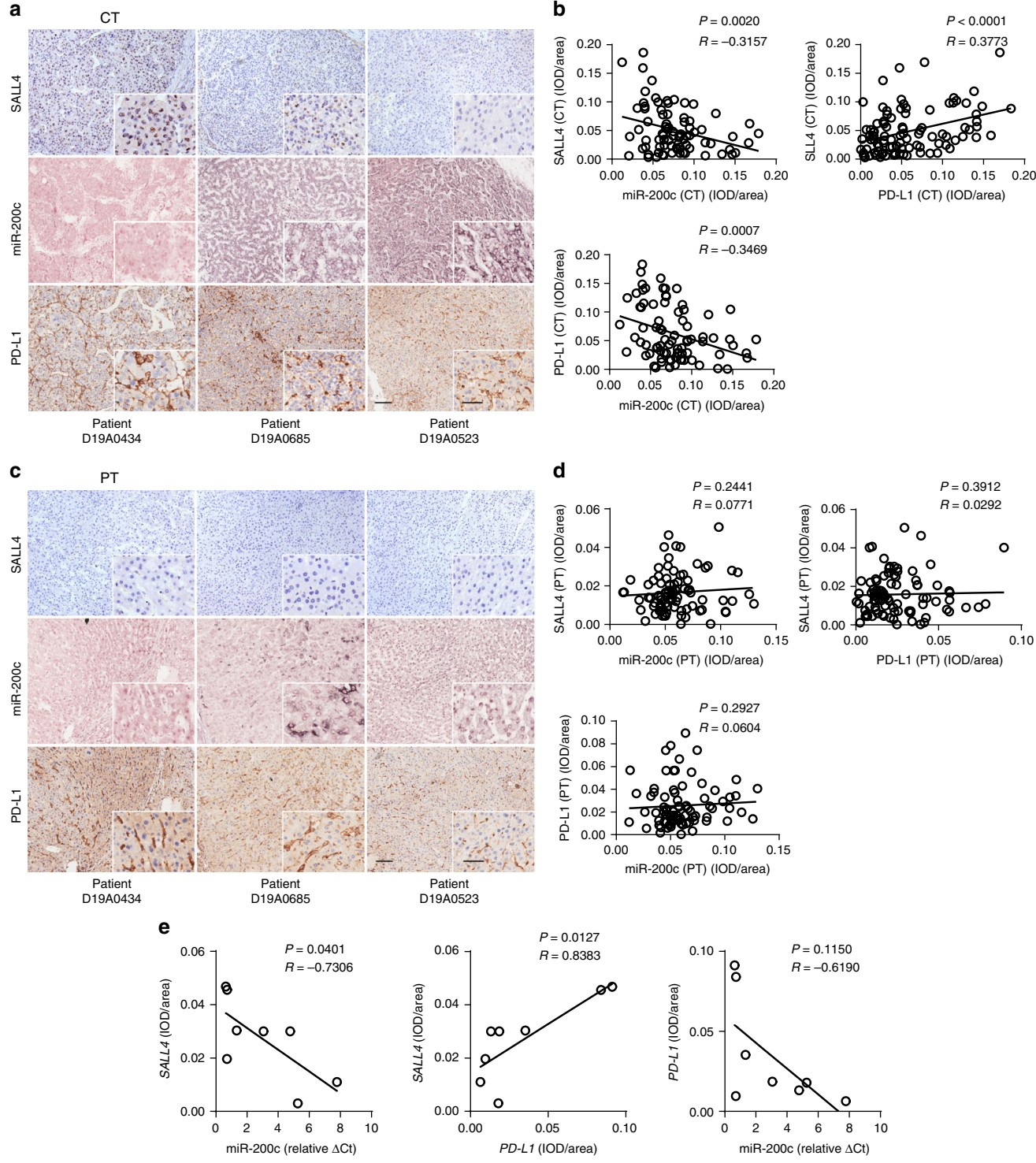

**Fig. 2** Correlations among SALL4, miR-200c, and PD-L1 in the center tumor regions of HCC. **a**, **c** Immunostaining of SALL4 and PD-L1 proteins and miR-200c in situ hybridization in the center tumor **a** and peritumor **c** regions of three representative HCC patients. Original magnifications: × 10, × 20; bar = 100 μm, 50 μm, respectively. **b**, **d** Pearson correlation analysis between the expression (IOD/area) of SALL4, PD-L1 and miR-200c in center tumor (**b**) and peritumor (**d**) regions of HCC. **e** Correlation analysis between *SALL4* and miR-200c (Relative ΔCt) (left), *SALL4* and *PD-L1* (middle), and *PD-L1* and miR-200c (right), from eight fresh tissue HCC samples. Spearman's correlation coefficients and *P* values are shown

$R = -0.3157$, Pearson's correlation coefficients) or miR-200c and PD-L1 ($P = 0.0007$, $R = -0.3469$, Pearson's correlation coefficients), accompanied with a positive correlation between SALL4 and PD-L1 expression ($P < 0.0001$, $R = 0.3773$, Pearson's correlation coefficients) (Fig. 2b); however, no statistical correlation among these molecules was identified in peritumor

regions ($P = 0.2441$, $R = 0.0771$; $P = 0.2927$, $R = 0.0604$; $P = 0.3921$, $R = 0.0292$, Pearson's correlation coefficients) (Fig. 2c, d). To further verify these two pairs of negative correlation between SALL4 and miR-200c or miR-200c and PD-L1, we investigated the expressions of miR-200c utilizing q-RT-PCR on eight fresh HCC samples. As expected, we

identified negative correlations between miR-200c expression and the integrated optical density (IOD)/area of either *SALL4* or *PD-L1* (Fig. 2e, left and middle).

To determine the specificity of miR-200c interacting with SALL4 or PD-L1, we examined the expression of miR-200a and miR-383 which might target *PD-L1* as shown in Fig. 1a, and analyzed their association with SALL4 and PD-L1 expression in HCC patients. We found that though miR-200a and miR-383 expressed higher in the center tumor regions than in peritumor regions (Supplementary Fig. 1a–c), neither disparities (Supplementary Fig. 1d–g) nor correlations (Supplementary Fig. 1h–n) of these two miRNAs with either PD-L1 or SALL4 expression from both the center tumor and peritumor regions were observed, except a positive correlation between PD-L1 and miR-200a in the peritumor regions (Supplementary Fig. 1o), indicating these negative correlations between miR-200 family and SALL4 or PD-L1 are more specific for miR-200c.

Interestingly, we observed a significant increase in SALL4, PD-L1, and miR-200c expression, not only in tumor tissue as already shown in Fig. 1b–e and Supplementary Fig. 2a, but also in liver cirrhosis ($n = 13$) (Supplementary Fig. 2a–d), suggesting chronic inflammation such as cirrhosis may stimulate expression of SALL4, PD-L1, and miR-200c. However, we did not find any significant disparity between miR-200c and PD-L1 or miR-200c and SALL4 (Supplementary Fig. 2e), indicating HCC microenvironment for those molecule expressions being different from that in cirrhosis.

**SALL4 and miR-200c correlate with overall survival**. Since the prognostic efficacy of PD-L1 has already been demonstrated in several human cancers, and the negative correlations of SALL4 to miR-200c and miR-200c to PD-L1 in HCC were identified by us as above, we evaluated the potential prognostic value of SALL4 and miR-200c in the center tumor regions. Statistical analysis revealed that a negative correlation between PD-L1 (integrated optical density (IOD)/area) and overall survival (OS) in the center tumor ($P < 0.0001$, $R = -0.4127$, Pearson's correlation coefficients) (Fig. 3a) but not in peritumor regions ($P = 0.2044$, $R = -0.0949$, Pearson's correlation coefficients) (Fig. 3b) has been identified. As expected, higher PD-L1 expression in the center tumor regions was significantly correlated with shorter OS of HCC patients ($P < 0.0001$, log-rank test) (Fig. 3c). In the same HCC group, a positive correlation of miR-200c expression to OS also existed in the center tumor ($P = 0.0007$, $R = 0.3613$, Pearson's correlation coefficients) and peritumor regions ($P = 0.0062$, $R = 0.2731$, Pearson's correlation coefficients) (Fig. 3d, e); and patients with higher miR-200c density had better prognoses in terms of OS ($P = 0.0307$, log-rank test) (Fig. 3f). Interestingly, similar to PD-L1, a negative correlation of SALL4 in the center tumor ($P < 0.0001$, $R = -0.3980$, Pearson's correlation coefficients) (Fig. 3g) but not peritumor regions ($P = 0.1227$, $R = -0.1340$, Pearson's correlation coefficients) (Fig. 3h) to OS was observed, demonstrating that patients with higher SALL4 expression in the center tumor regions manifested shorter OS ($P < 0.0001$, log-rank test) (Fig. 3i). Moreover, a comprehensive analysis showed that patients with lower SALL4 and higher miR-200c (Fig. 3j), or with lower SALL4 and lower PD-L1 (Fig. 3k), or with lower PD-L1 and higher miR-200c (Fig. 3l) had significantly longer OS. These findings demonstrate the prognostic value of SALL4 or miR-200c in center tumor regions and, therefore, might possibly serve as a promising predictor of survival in HCC. The correlations among SALL4, miR-200c, and PD-L1 expression to OS were further identified utilizing Cox regression with time-to-event outcome analysis (Supplementary Table 2), showing PD-L1 ($P < 0.0001$), SALL4 ($P < 0.0001$), and miR-200c ($P = 0.0020$) in tumors

significantly influenced OS, respectively. We then performed Cox multivariate regression analysis by adding tumor metastasis, TNM stages, tumor volume, SALL4, miR-200c, and PD-L1 into a model. Interestingly, SALL4 and TNM stages have the most significant association with OS (HR: 2.16, 1.71, respectively; $P = 0.0407$, 0.0264; Supplementary Table 3).

**miR-200c down-regulates HBV-induced PD-L1 expression**. Since HBV chronic infection is a key factor in hepatocellular carcinogenesis, to examine the underlying mechanism, we analyzed the levels of PD-L1 on HBV[+]human liver cell line HLCZ01 cells (HBV[+]HLCZ01)[20] that were infected with HBV particles from the supernatant of HepG2.2.15 cells overnight, and found that the expression of PD-L1 in HBV[+]HLCZ01 cells was higher than that in uninfected HLCZ01 cells at both mRNA and protein levels (Supplementary Fig. 3a). To address the possibility that HBV affects the expression of PD-L1 by hepatocytes in vivo, we utilized HBV-persistent mice via hydrodynamic (h.d.) injection of pAAV/HBV1.2 plasmid into C57BL/6 mice[21]. We noted that the expression of PD-L1 by primary hepatocytes increased in HBV-persistent mice than in HBV-negative mice (Supplementary Fig. 3b, c) and, moreover, the level of PD-L1 on primary hepatocytes positively correlated with the serum levels of HBsAg, HBeAg, and HBV-DNA (Supplementary Fig. 3d, e), indicating that HBV-persistence might promote PD-L1 expression by hepatocytes.

We examined PD-L1 expression of HBV[+]HLCZ01 cells transfected with 50 nM synthetic miRNA mimics or inhibitors of miR-200c, miR-200a, and miR-383, which were all reduced in HBV[+]HLCZ01 cells, with a possibility of targeting the 3′-UTR of *PD-L1* (miRBase and miRNA database) (Fig. 1a). PD-L1 was significantly reduced by miR-200c mimics and markedly enhanced by miR-200c inhibitors, while neither miR-200a nor miR-383 mimics significantly altered the PD-L1 level on HBV[+]HLCZ01 cells (Fig. 4a, Supplementary Fig. 4). These effects of miR-200c mimics and inhibitors on PD-L1 expression were dose dependent (Fig. 4b). To further confirm the direct targeting of miR-200c to the 3′-UTR of *PD-L1*, we constructed a luciferase reporter vector that contained the 3′-UTR of *PD-L1* with the putative miR-200c binding site (pMIR-Reporter-*PDL*-UTR), and co-transfected HBV[+]HLCZ01 cells with pMIR-Reporter-*PDL*-UTR, together with miR-200c mimics, inhibitor, or nonspecific control, followed by an assessment of luciferase activity 36 h after transfection. As shown in Fig. 4c, the luciferase activity was significantly decreased by the miR-200c mimics, but markedly enhanced by the miR-200c inhibitor. Taken together, our results indicated that miR-200c directly targets *PD-L1* at the 3′-UTR.

To explore whether miR-200c antagonizes HBV to affect expression of PD-L1, we infected HLCZ01 cells with HBV particles (co-cultured with a supernatant of HepG2.2.15 cells) and examined the relative expression of miR-200c. We found that miR-200c expression was significantly inhibited in HBV[+]HLCZ01 cells (Fig. 4d), indicating that HBV inhibited miR-200c expression. On the other hand, we transfected miR-200c mimics into HBV[+]HLCZ01 cells, and found that miR-200c almost completely abrogated HBV-mediated up-regulation of PD-L1 expression, displayed by an almost equal PD-L1 level to that of untreated HLCZ01 cells (Fig. 4e), indicating that miR-200c impedes HBV-mediated PD-L1 expression. Similar results were also obtained from another in vitro HBV infection model using HepaRG cell line (Fig. 4f). We constructed a miR-200c over-expression vector (pcDNA-miR-200c), which was hydrodynamically injected into HBV-persistent mice. As expected, miR-200c in HBV[+]hepatocytes, isolated from HBV-persistent mice, significantly decreased than that in HBV[−]mice (Fig. 4g).

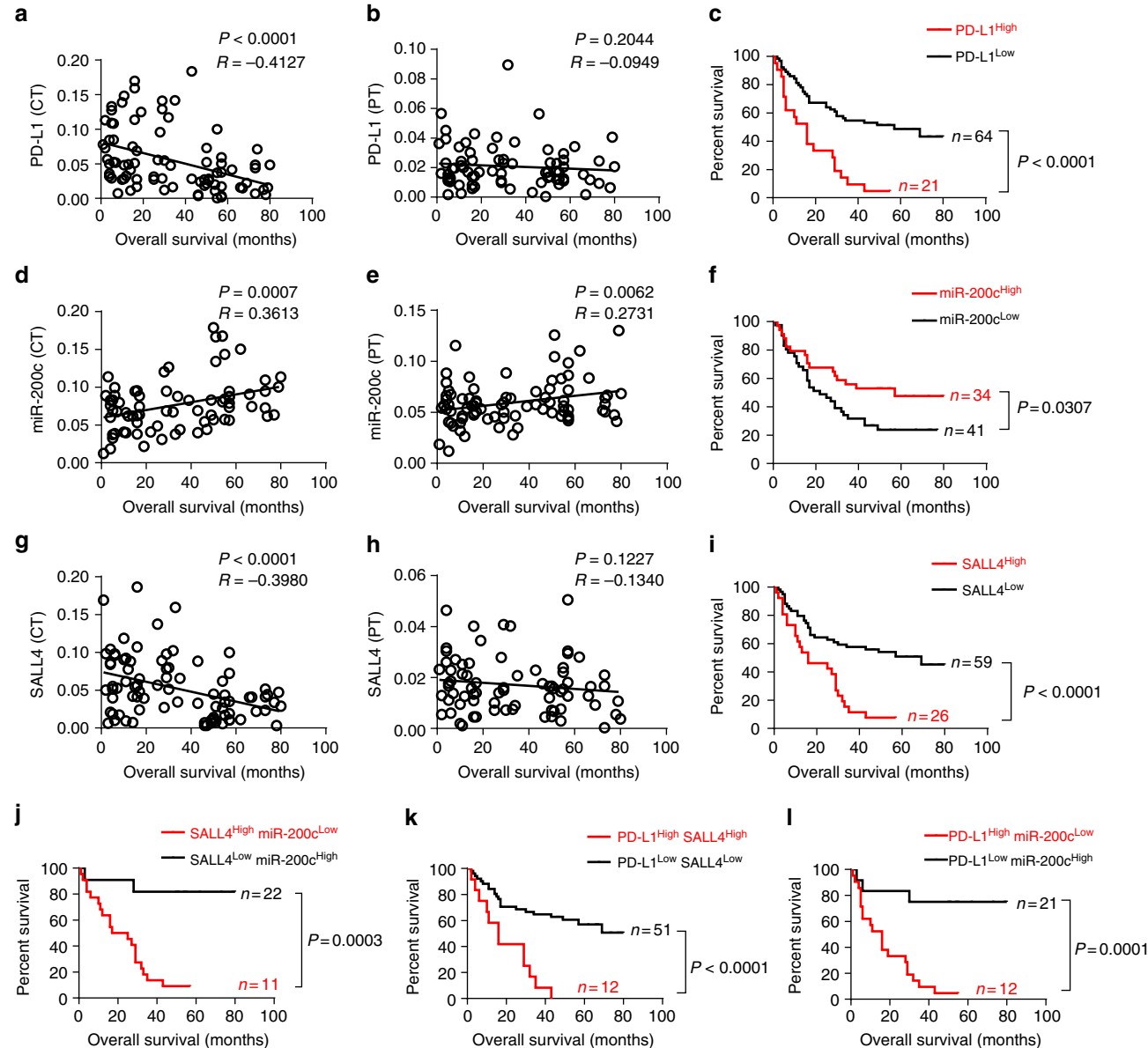

**Fig. 3** Increased overall survival in patients with lower SALL4, higher miR-200c and lower PD-L1. **a**, **b** Correlation of PD-L1 expression in the center tumor and peritumor regions to overall survival (OS). **c** OS in patients with PD-L1 expression in the center tumor regions (log-rank test). **d**, **e** Correlation of miR-200c expression in center tumor and peritumor regions to OS. **f** OS in patients with miR-200c expression in the center tumor regions (log-rank test). **g**, **h** Correlation of SALL4 expression in the center tumor and peritumor regions to OS. **i** OS in patients with SALL4 expression in center tumor regions (log-rank test). **j–l** OS in patients concurrently with SALL4 and miR-200c, or SALL4 and PD-L1, or miR-200c and PD-L1 (log-rank test). Pearson's correlation coefficients (r) and P values are shown in Fig. a, b, d, e, g, and h

Conversely, miR-200c over-expression significantly attenuated HBV-induced up-regulation of PD-L1 expression on hepatocytes (Fig. 4h). These data demonstrate that there exists an antagonism between HBV and miR-200c during PD-L1 expression within HBV-infected hepatocytes.

**miR-200c reverses CD8+ T cell exhaustion in HBV+ mice.** Similar to previous clinical and mouse studies[7,8,21], expression of PD-1 and other inhibitory receptors (Tim-3, LAG-3, CD160, BTLA, etc.) by hepatic CD8+T cells in HBV+ mice were higher than that in HBV−mice, and accompanied with lower percentages of CD69+ (activated) or IFN-γ+ (effector) CD8+T cells (Supplementary Fig. 5a−d), indicating CD8+T cell became exhausted in HBV+ mice in our setting, similar to that we reported previously[21]. Additionally, after blocking Pd-l1/Pd-1 pathway with an anti-Pd-l1

antibody, the percentages of CD69+CD8+ and IFN-γ+CD8+T cells were significantly recovered in HBV-persistent mice (Supplementary Fig. 5e). To further confirm that the elevated Pd-l1 on hepatocytes could impair the activation of hepatic CD8+ T cells, we constructed mouse Pd-l1 over-expression vector pEGFP-mPd-l1, which were h.d. injected into mice. We confirmed higher Pd-l1 expression by hepatocytes in mice injected with pEGFP-mPd-l1, and found these mice accompanied with much lower hepatic CD69+CD8+ and IFN-γ+CD8+T cells and higher apoptotic CD8+T cells (Supplementary Fig. 5f–h). These results indicate that the elevated Pd-l1 on hepatocytes induces exhaustion of CD8+T cells in HBV-persistent mice.

We then explored the role of miR-200c in HBV-induced CD8+T exhaustion. As shown in Fig. 5a−d, the percentages of hepatic CD69+CD8+, IFN-γ+CD8+, and CD107a+CD8+T cells,

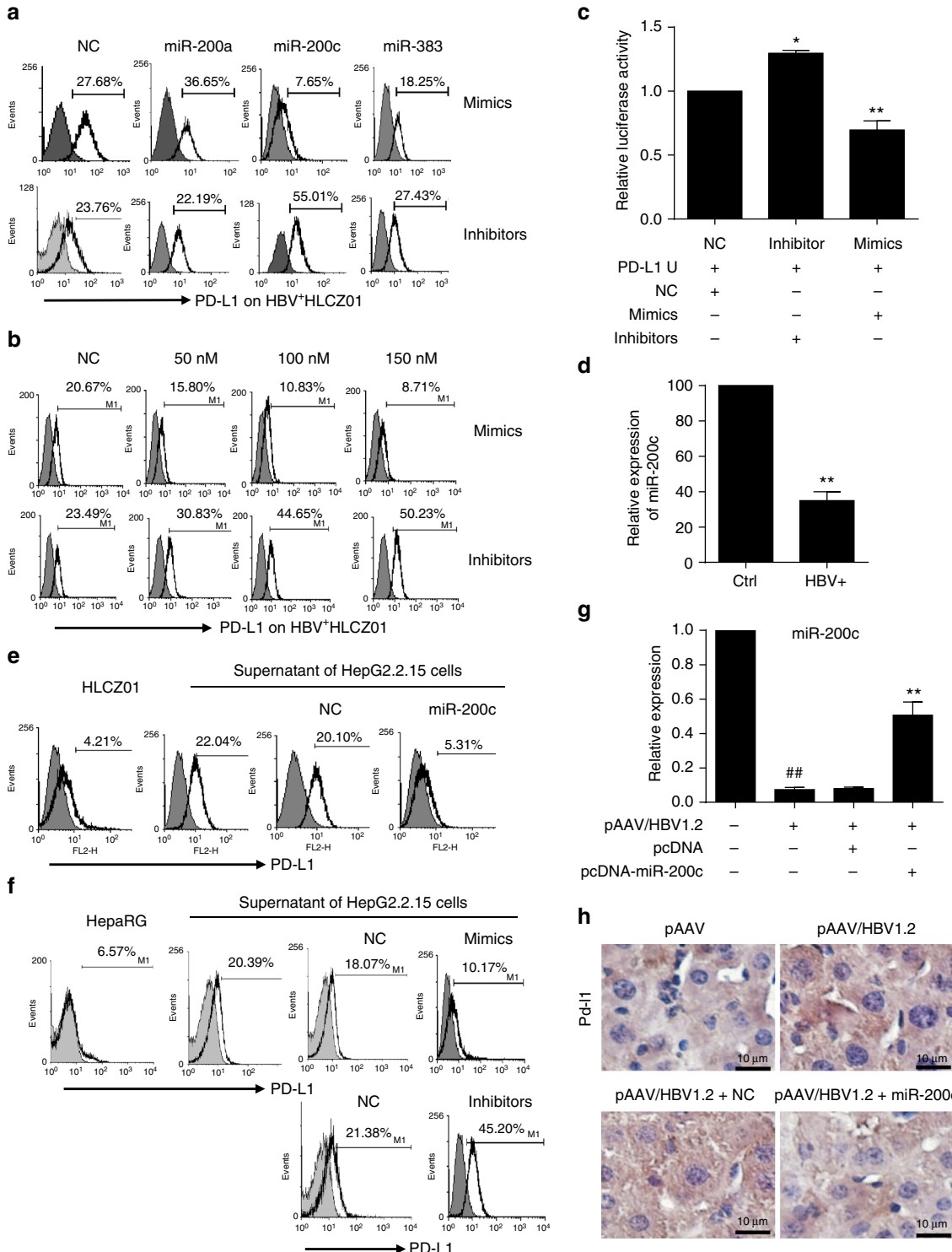

**Fig. 4** miR-200c down-regulates PD-L1 on hepatoma cells. **a** PD-L1 expression on human HBV+ hepatoma cells. 50 nM miR-200a, miR-200c, and miR-383 mimics or inhibitors were transfected into HBV+HLCZ01 cells; 24 h later, PD-L1 was detected by FACS. **b** Indicated doses of miR-200c mimics and inhibitors were transfected into HBV+HLCZ01 cells; 24 h later, PD-L1 was detected by FACS. **c** HBV+HLCZ01 cells were transfected with pMIR-Reporter-*PD-L1*-3′-UTR and pRL-TK, together with 100 nM of miR-200c mimics, inhibitors, or NC for 36 h. Renilla luciferase activity was normalized to Firefly luciferase activity. **d** Relative expression of miR-200c in HLCZ01 and HBV+HLCZ01 cells. **e** PD-L1 expression after miR-200c or NC was transfected into HBV+HLCZ01 cells by FACS. **f** miR-200c mimics and inhibitors were transfected into HBV+HepaRG cells; 24 h later, PD-L1 was detected by FACS. **g** Influence of HBV in miR-200c expression in HBV-persistent mice. pAAV/HBV1.2 plasmid together with pcDNA-miR-200c or pcDNA-empty was hydrodynamically injected into C57BL/6 mice; four weeks later, miR-200c expression in hepatocytes was detected by q-RT-PCR. **h** Influence of miR-200c in Pd-l1 expression in liver tissue of HBV-persistent mice. Pd-l1 was analyzed by IHC. Data are representative results from three independent experiments. Significant differences calculated using unpaired two-tailed Student's $t$-tests. *$P < 0.05$, **$P < 0.01$, versus NC or HBV− control (**c**, **d**); ##$P < 0.01$, versus HBV− mice. **$P < 0.01$, versus pAAV/HBV1.2 with or without pcDNA injection (**g**)

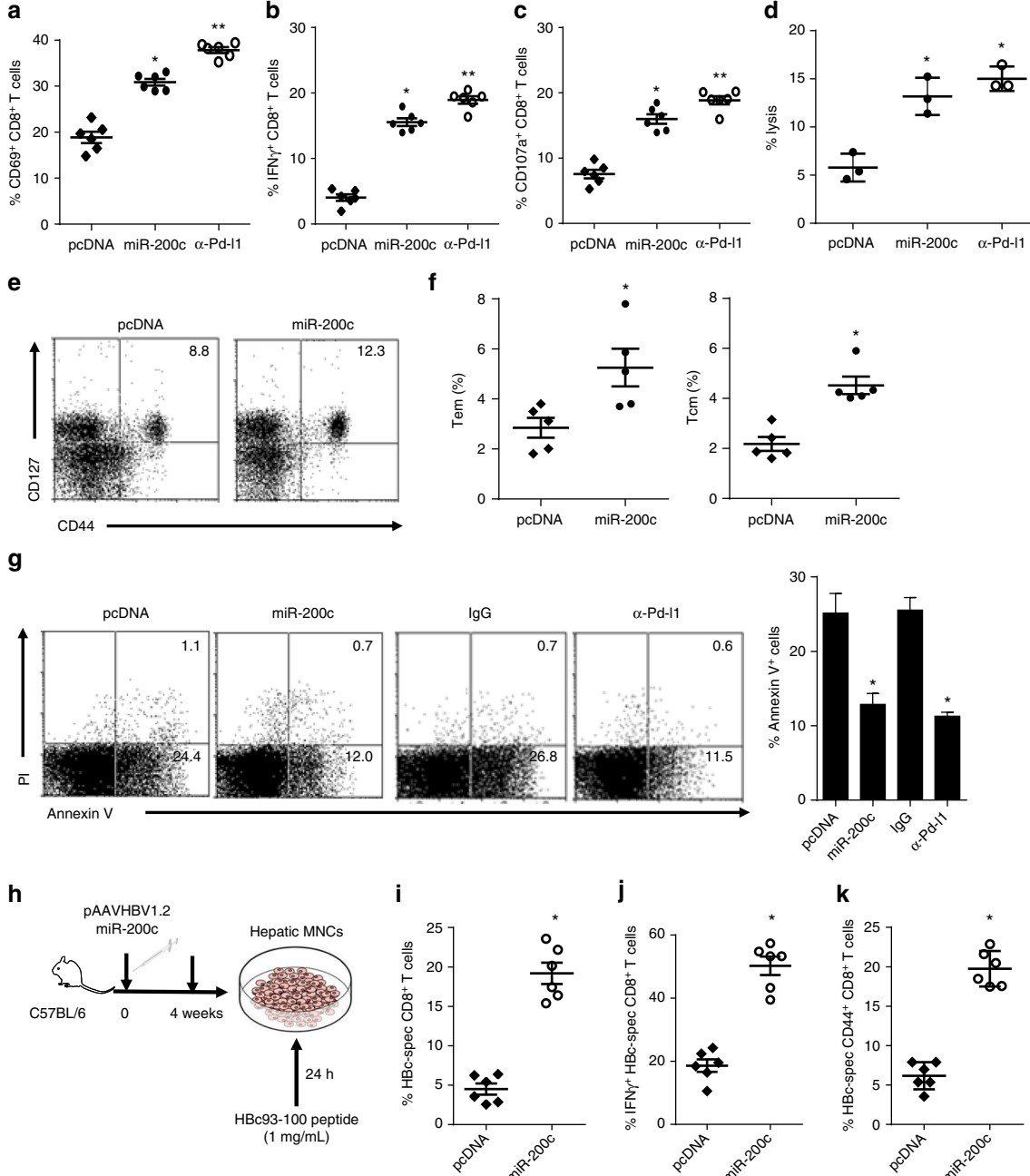

**Fig. 5** miR-200c reverses CD8+T cell exhaustion in HBV-persistent mice. **a, b** Activation and IFN-γ-producing CD8+ T cells in HBV-persistent mice by miR-200c over-expression. Plasmids containing pAAV/HBV1.2, together with plasmids containing pcDNA-miR-200c or pcDNA, were hydrodynamically injected into C57BL/6 mice, or treated with anti-Pd-1 Ab, respectively; four weeks later, percentages of CD69+CD8+ and IFN-γ+CD8+ T cells in all CD8+ T cells were detected by FACS. **c, d** Cytolytic function of CD8+T cells in HBV-persistent mice by miR-200c over-expression. Percentages of CD107a+ CD8+ T cells in all CD8+ T cells and cytolysis by CD8+T cells were detected by FACS. **e, f** Memory CD8+ T cell in HBV-persistent mice by miR-200c over-expression. CD44+CD127+memory CD8+ T cells, CD44hiCD62Llow effector CD8+ T cells (Tem) and CD44hiCD62Lhi central memory CD8+ T cells (Tcm) in total CD8+ T cells were detected by FACS. **g** The apoptosis of CD8+ T cells was analyzed by double staining of Annexin V and PI. **h–k** Hepatic CD8+T cells from HBV-persistent mice in response to HBc93-100 stimulation in vitro. Hepatic lymphocytes from HBV-persistent mice, treated with or without miR-200c over-expression as above, were separated and stimulated with HBc93-100 peptide-loaded monocytes (1 mg/mL) for 24 h. The percentages of HBc-specific CD8+ T cells (**i**) and HBc-specific IFN-γ+CD8+ T cells in total CD8+ T cells (**j**), and HBc-specific CD44+CD127+memory CD8+ T cells (**k**) were detected by FACS. Data are representative of three independent experiments with six mice per group. Significant differences calculated using unpaired two-tailed Student's t-tests. *P < 0.05, **P < 0.01, versus pcDNA control

and cytolysis against target cells were considerably higher after liver-specific miR-200c over-expressed in mice, approximating the levels attained by Pd-l1 blocking, along with higher CD44+CD127+memory, CD44hiCD62Llow effector memory, CD44hiCD62Lhi central memory CD8+T cells (Fig. 5e, f,

Supplementary Fig. 4) and lower apoptotic CD8+T (Fig. 5g), demonstrating a reversal of functional exhaustion of CD8+T cell was obtained by hepatic miR-200c over-expression in vivo. Importantly, examined in response to HBc93-100 peptide-pulsed autologous monocytes in vitro (Fig. 5h), hepatic lymphocytes

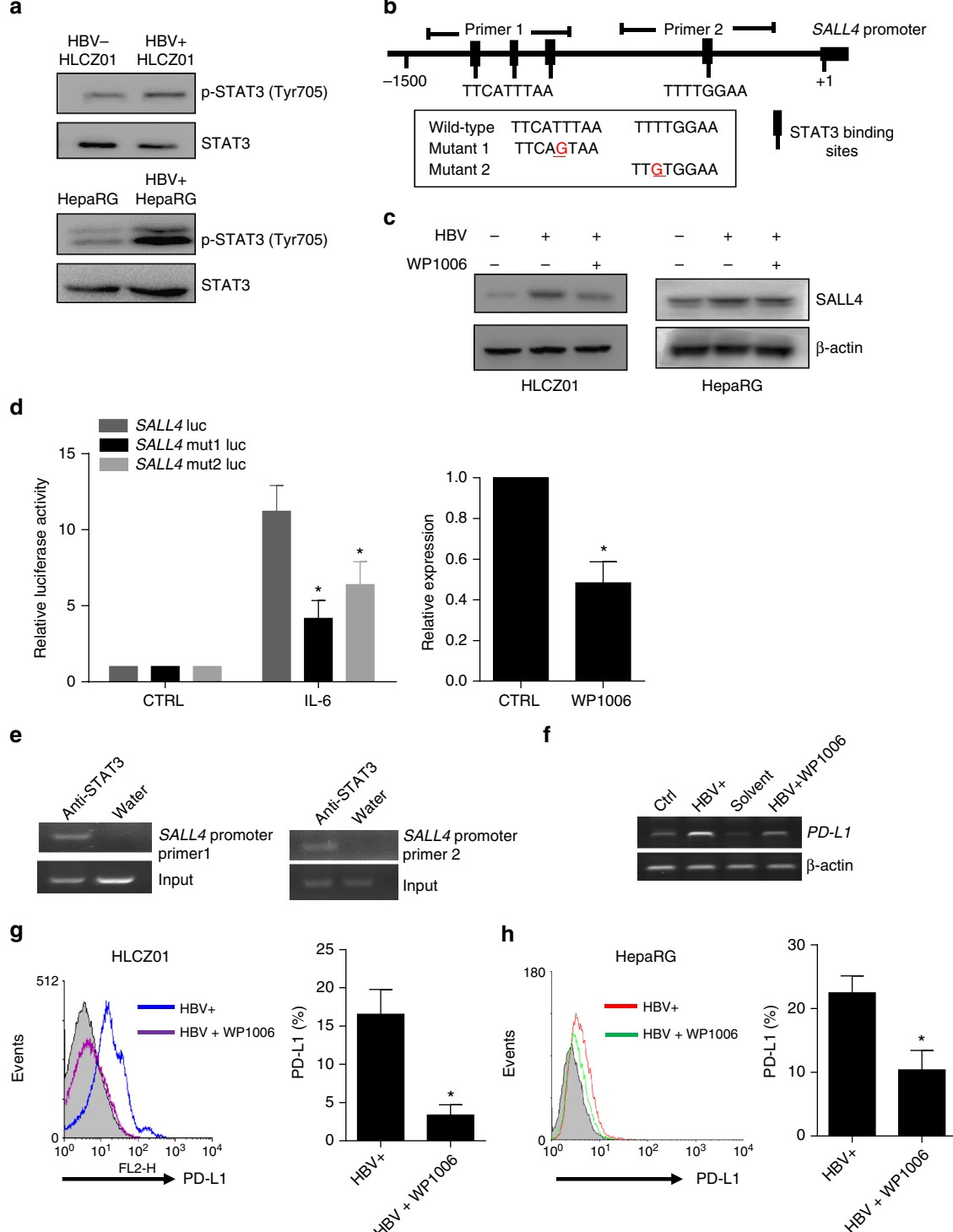

**Fig. 6** HBV triggers SALL4 expression via STAT3 activation. **a** STAT3 activation was assayed in HBV⁻ and HBV⁺ Hepatoma cells. HBV-infected or uninfected HLCZ01 (HBV⁺HLCZ01 and HLCZ01) cells, and HBV⁺HepaRG cells and HepaRG were used to detect STAT3 activation by western blot. **b** Upstream promoter region of the human *SALL4* showing putative STAT3-binding sites. A schematic diagram of the -1500/ + 20 human *SALL4*-luciferase construct is shown, and the sequence of the point mutations also indicated. Schematic representative primer 1 and primer 2 specific for the *SALL4* promoter used in this study were indicated. **c** SALL4 protein was determined in HBV⁺HLCZ01 cells and HBV⁺HepaRG cells treated with/without STAT3 inhibitor WP1006. **d** HBV⁺HLCZ01 cells were transfected with reporter plasmids containing *SALL4* promoter or *SALL4* mutant (mut 1 and mut 2), together with IL-6 or WP1006 treatment. The Renilla luciferase activity was normalized to Firefly luciferase activity. **e** ChIP with STAT3 antibody shows binding of STAT3 to the *SALL4* promoter in HBV-infected HLCZ01 cells. **f**, **g** HBV⁺HLCZ01 cells were treated with/without STAT3 inhibitor WP1006; 24 h later, the mRNA level (**f**) and surface protein expression (**g**) of PD-L1 were determined via q-RT-PCR and FACS, respectively. **h** HBV⁺HepaRG cells were treated with/without STAT3 inhibitor WP1006; 24 h later, surface protein expression of PD-L1 was determined via FACS. Data are representative of three independent experiments. Significant differences calculated using unpaired two-tailed Student's *t*-tests. *$P < 0.05$

from miR-200c over-expressed mice contained more HBc-specific CD8[+]T cells (Fig. 5i) and in particular, more HBc-specific IFN-γ [+]CD8[+]T cells (Fig. 5j) and a greater number of HBc-specific memory CD8[+]T cells (CD44[+]CD127[+]) (Fig. 5k), suggesting that miR-200c could promote HBV-specific CD8[+]T cell function in vivo.

Meanwhile, notably, miR-200c over-expression significantly decreased the serum levels of HBV DNA and HBsAg, and the tissue levels of HBsAg and HBcAg in the livers of HBV-persistent mice (Supplementary Fig. 6a−c), implying the recovery of antiviral CD8[+]T cell function by miR-200c-mediated PD-L1 repression facilitates HBV inhibition by antiviral immunity. Also,

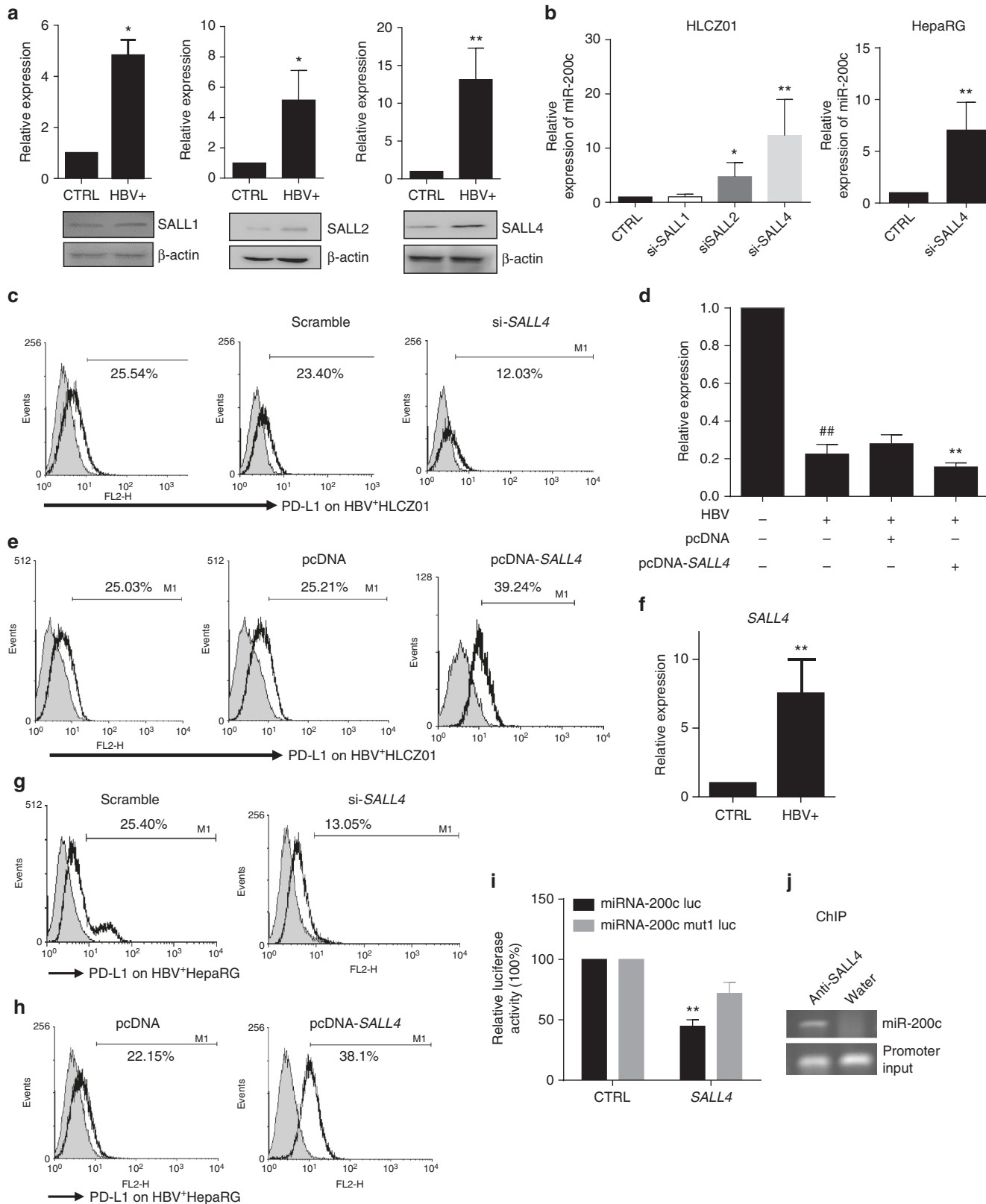

we h.d. pre-injected mice with pcDNA-miR-200 vector, and one week later, injected pAAV/HBV1.2 to establish HBV-persistent mice. We found the serum levels of HBV DNA and HBsAg in mice with a miR-200c pre-injection became significantly low (Supplementary Fig. 6d), suggesting that miR-200c over-expression could interfere with HBV persistence via improving T cell function.

**HBV down-regulates miR-200c via STAT3-induction of SALL4.** Furthermore, we explored the relationship between HBV infection and miR-200c. We found that neither miR-200c mimics nor miR-200c inhibitors directly impaired the expression of HBsAg and HBcAg in hepatocytes in vitro. However, as described in Fig. 4d, HBV infection significantly inhibited miR-200c expression, indicating that HBV infection could intrinsically decrease miR-200c transcription. It was noted that STAT3 mediated the up-regulation of PD-L1 in monocytes, plasmacytoid dendritic cells (pDCs) and myeloid-suppressor cells infiltrating liver metastases[22,23]. On comparing the phosphorylation level of STAT3 in HBV+ and HBV− hepatoma cell lines, we found that the HBV promoted STAT3 activation (Fig. 6a, Supplementary Fig. 7). Of special interest, -1.5-kb SALL4 promoter region revealed four consensus sequences for STAT3 binding using DNA sequence analysis (Fig. 6b)[24]. We found that the STAT3 inhibitor WP1006 significantly attenuated HBV-induced SALL4 expression in HLCZ01 and HepaRG cells (Fig. 6c, Supplementary Fig. 7). To confirm the role of STAT3 on the regulation of SALL4 expression, we constructed luciferase reporter vectors containing putative SALL4 promoters or mutated promoters. HLCZ01 cells were transfected with reporter vectors and simultaneously treated by IL-6, a STAT3-activating cytokine. As shown in Fig. 6d, IL-6 stimulation significantly enhanced the activity of wild-type but not mutant promoter of SALL4, possibly through STAT3 activation since luciferase activity of SALL4 promoter was significantly decreased if STAT3 activation was inhibited with WP1006 (Fig. 6d, right). Applying chromatin immunoprecipitation (ChIP), we found that endogenous STAT3 could directly bind to the promoter region of SALL4 (Fig. 6e, Supplementary Fig. 7), suggesting that HBV persistence might down-regulate miR-200c via STAT3-induced transcriptional repressor SALL4. As expected, WP1006 treatment significantly decreased PD-L1 expression by HBV+HLCZ01 cells at both mRNA and protein levels (Fig. 6f−g, Supplementary Fig. 7). Similar results were obtained from HBV-infected HepaRG cells (Fig. 6h).

Next, we tried to investigate how SALL4 regulates miR-200c transcription. Through bioinformatic analysis, we found that there are three putative binding sites of the SALL family within the promoter region of miR-200c (Fig. 1a). Importantly, HBV infection significantly promoted the expression of SALL family, particularly SALL4, by HLCZ01 cells at both mRNA (up) and protein (below) levels (Fig. 7a, Supplementary Fig. 8). Therefore, we explored whether SALL4 contributes to the regulation of miR-200c transcription. As shown in Fig. 7b, c, silencing SALL4 markedly enhanced miR-200c expression and led to decrease in

HBV-induced PD-L1 expression; whereas over-expression of SALL4 significantly reduced miR-200c transcription and augmented PD-L1 expression (Fig. 7d,e). Furthermore, we observed SALL4 was induced in HBV+HepaRG cells (Fig. 7f). Similarly, silencing SALL4 led to augmented miR-200c transcription (Fig. 7b, right) and decreased PD-L1 expression (Fig. 7g), while over-expression of SALL4 resulted in PD-L1 up-regulation (Fig. 7h). Luciferase reporter and ChIP assay further confirmed that SALL4 suppressed the promoter activity of miR-200c (Fig. 7i) and could directly bind to the promoter region of miR-200c (Fig. 7j, Supplementary Fig. 8). Together, these results clearly demonstrated that HBV persistence promotes the expression of the transcriptional repressor SALL4, which then directly suppresses miR-200c expression by binding to the promoter region.

**Silencing Sall4 hinders HBV replication and CTL exhaustion.** To further confirm the role of SALL4 in promoting HBV persistence and HCC progression in vivo, we silence liver Sall4 expression to observe the influence on HBV infection or persistence in HBV-persistent mice (Fig. 8a, b). Sall4 and Pd-l1 expression was higher while miR-200c transcription was lower in HBV-persistent mice than those in HBV− mice; while silencing Sall4 significantly enhanced miR-200c transcription and decreased Pd-l1 expression (Fig. 8c, d). Importantly, silencing Sall4 markedly augmented the percentages of IFN-γ+ CD8+T cells, reduced PD-1 expression on CD8+T cells, and suppressed CD8+T cell apoptosis (Fig. 8e), along with the decreased levels of HBsAg and HBeAg in HBV-persistent mice (Fig. 8f).

We further confirmed reactivation of SALL4 in HCC cells in association with CD8+T cell exhaustion in patients with HBV+HCC. We examined the levels of SALL4 expression, PD-1 expression, and CD8+T cell infiltration in the peritumor and center tumor regions of fresh biopsies of HCC patients by immunofluorescent staining. We found that SALL4 expression was much higher in the center tumor regions than in peritumor regions, while the content of tumor infiltrating CD8+T cells in the center tumor regions were much lower than those in peritumor regions (Fig. 9a); though the number of total tumor infiltrating CD8+T cells in the center tumor regions was decreased, which is consistent with our former finding[25], the content of PD-1+CD8+T cells was much higher in the center tumor regions than those in peritumor regions (Fig. 9a). Further, statistics analysis showed that there was a positive correlation between SALL4 expression and tumor infiltrating PD-1+ CD8+T cells (Fig. 9b), raising a possibility that hepatocyte SALL4 involves PD-1+CD8+T cell exhaustion via expressing PD-L1 during HBV+HCC progression.

## Discussion

Recent reports indicate that CD8+T cells are dysfunctional and exhausted, characterized by high PD-1 expression and low IFN-γ and TNF secretion in HBV-related HCC[2,4,7]. The elevated

**Fig. 7** HBV-STAT3-SALL4 axis inhibits miR-200c transcription. **a** The mRNA and protein levels of SALL1, SALL2 and SALL4 of HBV+HLCZ01 cells were determined by q-RT-PCR (upper) and western Blot (below). **b** miR-200c expression was determined in HBV+HLCZ01 cells transfected with si-SALL1, si-SALL2, and si-SALL4 (left), miR-200c expression was determined in HBV+HepaRG cells transfected with si-SALL4 (right). **c** PD-L1 expression was assayed by FACS in HBV+HLCZ01 cells transfected with si-SALL4. **d** miR-200c expression was determined in HBV+HLCZ01 cells transfected with SALL4 over-expressing vector. **e** PD-L1 expression was assayed in SALL4 over-expressed HBV+HLCZ01 cells. **f** The mRNA level of SALL4 in HBV+HepaRG cells was determined by q-RT-PCR. **g** PD-L1 expression was assayed by FACS in HBV+HepaRG cells transfected with si-SALL4. **h** PD-L1 expression was assayed by FACS in HBV+HepaRG cells transfected with SALL4 over-expressing vector. **i** The activity of the miR-200c promoter reporter was assayed. **j** ChIP with SALL4 antibody showed binding of SALL4 to the miR-200c promoter in the HBV-infected H LCZ01 cells. Data are representative of three independent experiments. Significant differences calculated using unpaired two-tailed Student's t-tests. *P < 0.05, **P < 0.01

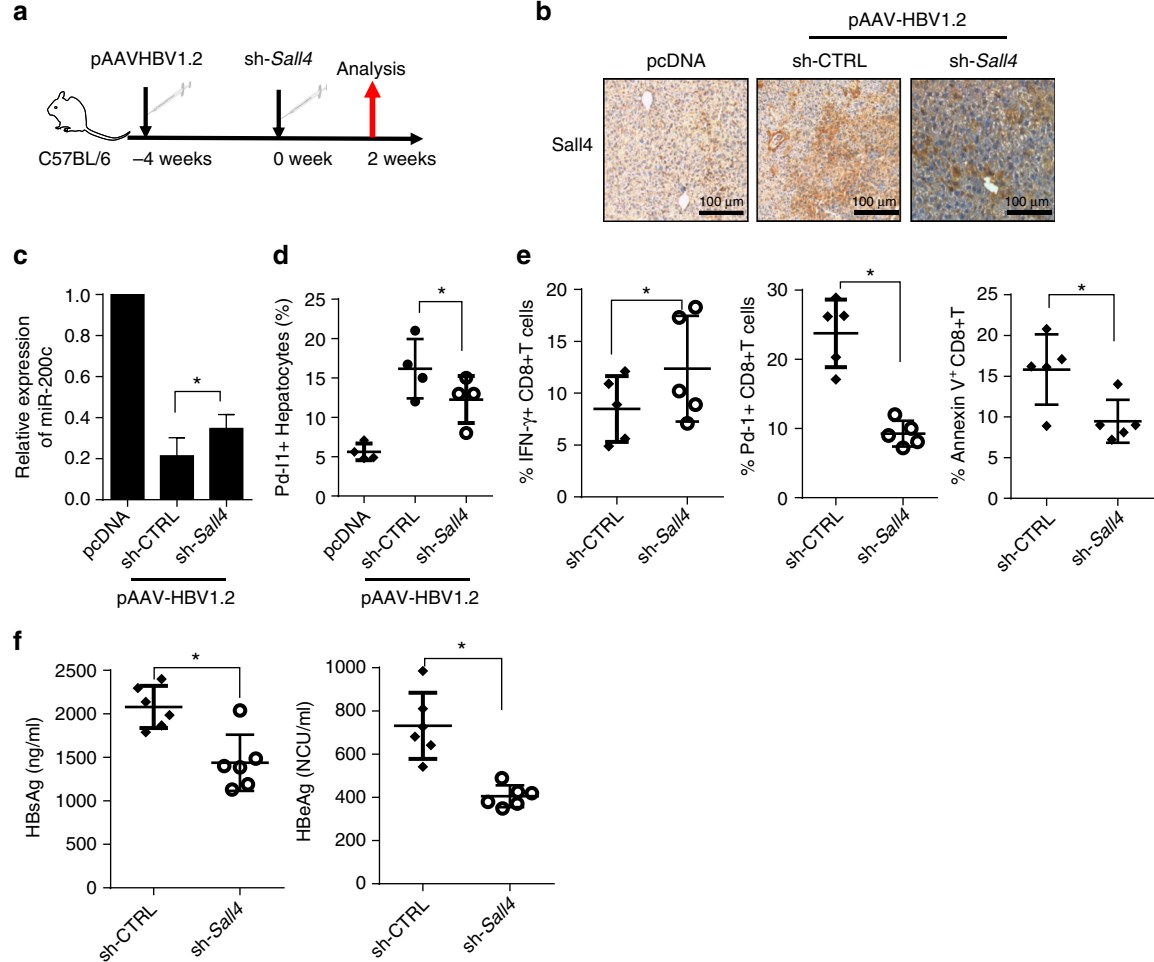

**Fig. 8** Silence of SALL4 hinders HBV persistence through miR-200c-PD-L1-T cell exhaustion. **a** Protocol of *Sall4* silence in HBV-persistent mice. pAAV/HBV1.2 was injected into C57BL/6 mice, and four weeks later, sh-*Sall4* or sh-ctrl was hydrodynamically injected into HBV-persistent mice, and then analyzed two weeks later. **b** The protein levels of Sall4 in liver were detected by IHC in HBV-persistent mice after sh-*Sall4* treatment. **c** miR-200c expression was determined by q-RT-PCR in primary hepatocytes from *Sall4*-silenced HBV-persistent mice. **d** Pd-l1 expression was assayed by FACS in the same cells as above. **e** The percentage of IFN-γ-, Pd-1-, and Annexin V-positive CD8+T cells was determined from liver of *Sall4*-silenced HBV-persistent mice. **f** The serum levels of HBsAg and HBeAg in *Sall4*-silenced HBV-persistent mice were detected by ELISA. Data are representative of three independent experiments. Significant differences calculated using unpaired two-tailed Student's *t*-tests. *$P < 0.05$

expression of PD-1 on lymphocytes is considered to be induced by its ligand PD-L1 on virus-infected cells or tumors, and the activation of PD-1/PD-L1 signal pathway may cause apoptosis or exhaustion of CD8+T cells. However, the underlying mechanism of PD-L1 expression, a critical process of T cell exhaustion induced by these target cells, remains unknown. Although it is extensively recognized that chronic viral infection such as HBV might induce T cell exhaustion via up-regulation of PD-L1[26,27], how HBV intrinsically up-regulates expression of PD-L1 remains elusive. In our clinical cohort, we found that both PD-L1 and SALL4 expression was negatively correlated with expression of miR-200c, respectively, in HBV-related HCC, and patients with high miR-200c and lower levels of SALL4 or PD-L1 manifested a significantly prolonged survival time. We further determined that miR-200c can directly target the 3′-UTR of *PD-L1*, and that miR-200c mimics could decrease the PD-L1 expression by HBV+hepatoma cells but miR-200c inhibitors could increase it by using a naturally HBV-infected HCC cell line, a recently established and solidly verified one[20]. Furthermore, over-expression of miR-200c could reverse the high expression of Pd-l1 and CD8+T cell exhaustion in HBV-persistent mouse, an immune uncompromised animal with hepatocyte-specific HBV carrier in adults.

Hence, we propose that miR-200c is a natural cellular intrinsic regulator to directly inhibit PD-L1 expression; however, HBV could suppress miR-200c transcription via reactivating host oncofetal protein SALL4, a transcriptional repressor, which is up-regulated by HBV-induced STAT3 activation and directly inhibits miR-200c synthesis, eventually leading to high expression of PD-L1 and subsequent CD8+T cell exhaustion (Fig. 9c). This represents a novel mechanism for HBV, or even other chronic oncoviruses, to escape from host immune surveillance. Therefore, we, for the first time from our knowledge, delineate the critical role of the reactivation of oncofetal protein SALL4 in negatively regulating miR-200c transcription.

*Sall4* is one of the key factors for maintenance of pluripotency and self-renewal of embryonic stem cells[14,15]. Interestingly, it is highly expressed in both murine and human fetal liver, and declines gradually during development and becomes silenced in adulthood[16,28]; *SALL4* is also recognized as an oncofetal gene that was first described in leukemia, and then in various types of cancers including HCC[16,29,30] and is proposed to have diagnostic value in several tumors[31–33]. The progenitor-like subtype of HCC with higher expression of SALL4 is reported to be associated with aggression and poor prognosis in clinic[16–18]. Thus, SALL4 is

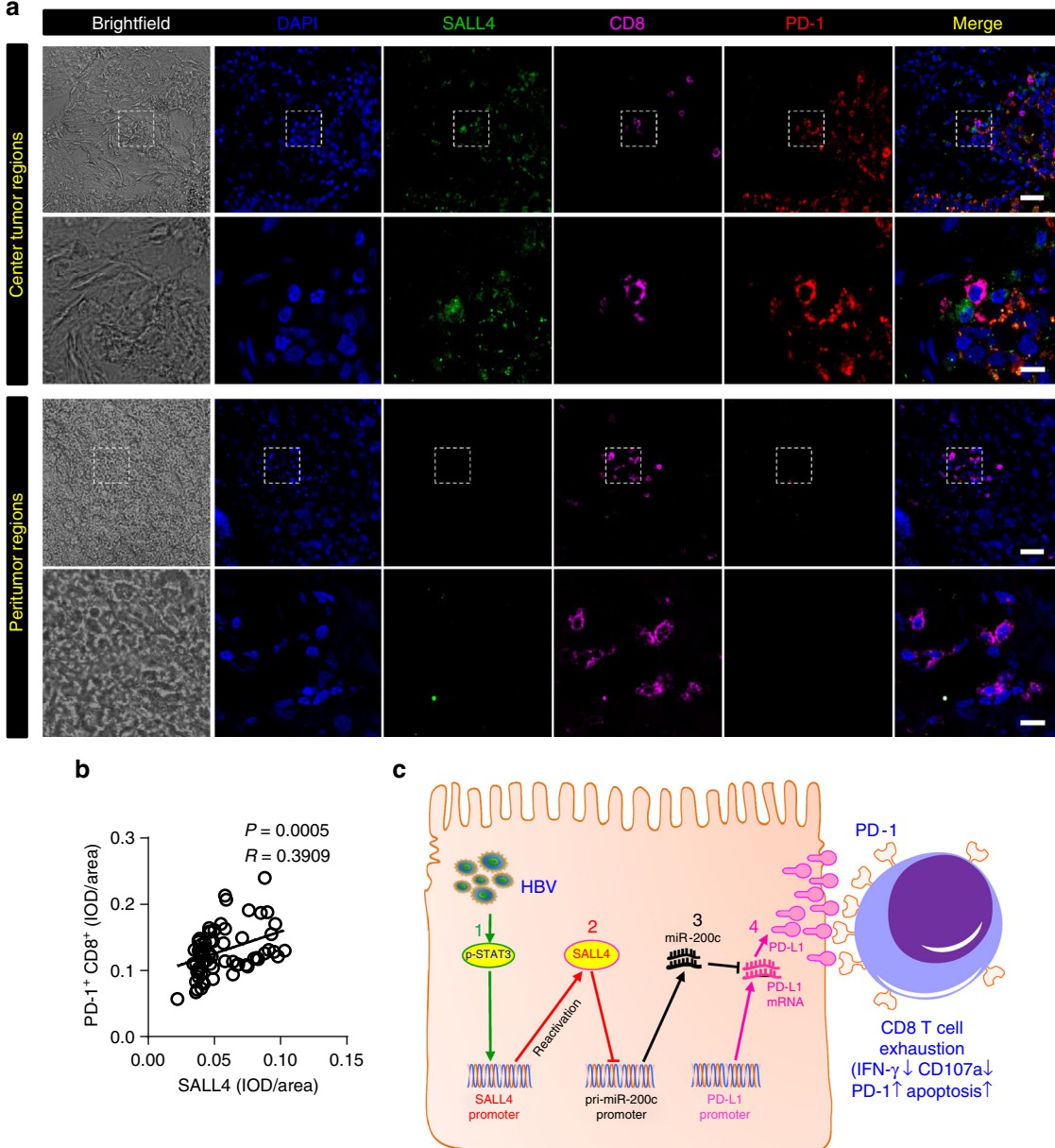

**Fig. 9** Positive correlation between SALL4 expression and tumor infiltrating PD-1⁺CD8⁺T cells in HCC. **a** Confocal immunofluorescence images of SALL4 (green), CD8 (magenta) and PD-1 (red) staining in the center tumor (top 2 row) and peritumor (bottom 2 row) regions of HCC. Original magnifications: × 20, × 63; scale bar represents 50 μm and 10 μm, respectively. **b** Correlation between the expression of SALL4 and PD-1⁺CD8⁺ cells in the center tumor regions of 75 confocal images from12 fresh tissue samples of HCC patients. Each patient has been taken more than six random perspectives. Pearson's correlation coefficients and $P$ values are shown. **c** The proposed model of HBV-pSTAT3-SALL4-miR-200c-PD-L1-T cell exhaustion axis. miR-200c is an intrinsic regulator to directly inhibit PD-L1 expression; however, HBV could suppress miR-200c transcription via reactivating oncofetal protein SALL4, a transcriptional repressor, up-regulated by HBV-induced STAT3 activation, eventually leading to higher expression of PD-L1 and subsequent CD8⁺ T cell exhaustion

currently considered as an emerging cancer candidate[16,34]. The expression of *Sall4* in fetal liver but not in adult liver, and importantly, re-expression in tumor liver led us to hypothesize that SALL4 might be an important mediator in HBV-induced PD-L1 expression which attributes to the immune tolerance for development and progression of HCC in adulthood. We found that HBV stimulates SALL4 production through activating STAT3, and then SALL4 directly suppresses miR-200c transcription by its transcriptional repressor function, subsequently leading to high PD-L1 expression since losing suppression by miR-200c (Fig. 9). Our results demonstrate a novel mechanism of HBV-induced PD-L1 regulation, under which

HBV works as a "knocker" to waken the "sleeping" SALL4 for counteracting miR-200c in adulthood. Therefore, SALL4 in HBV-pSTAT3-SALL4-miR-200c-PD-L1 axis possibly works as a repressor of repressor miR-200c in regulating PD-L1 expression.

Although SALL4 was recently reported to over-express in various cancers and to regulate cell proliferation, apoptosis, migration/invasion, and stemness by targeting a variety of genes, there is no report about SALL4 regulation of host miRNA transcription. This study is the first description that SALL4 directly targets and suppresses miR-200c, an epigenetically negative regulator for PD-L1 expression as described in this study, and finally results in T cell exhaustion and immune evasion via SALL4-miR-

200c-PD-L1 axis. Also, it is the first time to delineate that HBV improves PD-L1 expression through STAT3-SALL4-miR-200c pathway, especially evidenced by using in vitro mimicking natural HBV infection system, in vivo HBV-persistent mouse model, and human HCC patients. Therefore, our result provides believable proofs to support that SALL4 acts as a critical factor in HBV-maintained immune tolerance and HCC progression, and to elucidate the role of chronic virus infection in CD8[+]T cell exhaustion and tumor progression via pSTAT3-SALL4-miR-200c-PD-L1 axis, a mechanism of immune tolerance from intrinsic (e.g., HBV-induced PD-L1 expression on hepatocytes) to extrinsic (e.g., hepatocyte PD-L1-induced T cell exhaustion) pathways.

Over-expression of coinhibitory molecules, such as CTLA-4, PD-1, and Tim-3, often results in the functional inhibition or exhaustion of T cells, reversing which by antibodies gives rise to the promising therapeutic modality in tumor, called checkpoint immunotherapy[35]. Accumulating data have demonstrated that PD-1 is critical in establishing peripheral tolerance and recognized as the primary cause of T cell exhaustion in HBV or HCV persistence and HCC, while PD-L1-mediated inhibition of PD-1[+]T cells plays a major role[7,36]. In this study, we found that the staining patterns of PD-L1 were highly correlated with HBsAg[+] HCC subtype, which is consistent with our observation in HBV[+]hepatocytes of HBV-persistent mice (Supplementary Fig. 3) and the report from Xie Z who showed the correlation of in situ PD-L1 expression to HBV load in patients with CHB[37]. No differences of PD-L1 staining pattern have been observed among HCC patients with different histologic and morphological subtypes. Further, we observed that late-stage HCC patients (TNM IV, pathology grading 3) maintained higher levels of PD-L1 compared with those of early stage HCC patients (TNM I, pathology grading 1) (Supplementary Table 4). For the expression pattern of PD-L1 in HBV[+]HCC patients, as shown in Figs 1, 2, and Supplementary Fig. S3, according to the morphology and size of PD-L1[+] cells, we observed that PD-L1 was mainly expressed on hepatocytes in the center tumor regions, while it mainly expressed on bone-marrow-derived cells in peritumor regions and in regions with liver cirrhosis. Recent studies have tried to explore the factors influencing PD-L1 expression, and demonstrated that IFN-γ promotes PD-L1 expression on macrophage and tumor cells in a STAT1- or PDK2-dependent manne[38,39]. IL-27 up-regulates PD-L1 expression on naïve CD4[+]T cells or liver pDCs in a STAT1 or STAT3-dependent manner[22,40]. Importantly, miR-200 is recently reported to target PD-L1, which could be antagonized by transcriptional repressor zinc-finger E-box binding homeobox 1 (ZEB1), in lung cancer[41]. However, until now, no report demonstrated how viruses such as HBV intrinsically up-regulate PD-L1 expression with clear molecular mechanisms. A recent paper proposed a unique genetic mechanism of PD-L1 regulation caused by structural variations commonly disrupting the 3′ region of the PD-L1 gene[42]. On reanalyzing these data we found that these structural variations occur more commonly in hematological malignancy but are uncommon in solid tumors. Also, the authors did not find any structural variation in the 3′UTRs of the PD-L1 gene from 371 samples with HCC, and there was no HBV integration within or near the PD-L1 gene in 118 samples of HBV-related HCC, suggesting that structural variations in 3′UTRs of the PD-L1 gene might at least not be the main regulatory mechanism for the high expression of PD-L1 in hepatocytes from HBV-related HCC. Here, we found HBV reactivates SALL4 which then improves HCC progress through STAT3-SALL4-miR-200c-PD-L1 pathway, especially strongly evidenced by HBV-persistent mouse model and human HCC patients.

Due to the critical role of coinhibitory signals in inducing T cell exhaustion, blocking coinhibitory molecules for treating chronic viral infection and cancer has presented a promising application prospect. A CTLA-4 blocking antibody (ipilimumab) has been approved by FDA in 2011 for the treatment of metastatic melanoma, followed in 2014 by the approval of two PD-1 blocking antibodies, Pembrolizumab and Nivolumab, for treatment of advanced melanoma, squamous non-small cell lung cancer, and renal cell carcinoma[43,44]. Combination strategies with different immune checkpoint inhibitors, such as anti-CTLA-4 and anti-PD-1, with different mechanisms of action have also been approved and are associated with higher response rates[44,45]. For treatment of HBV infection and HCC, blocking CTLA-4 enhances expansion of IFN-γ[+]CD8[+]T cells in both peripheral and intrahepatic compartments. The blockage of Tim-3 promoted cytokine secretion during HBV infection both in patients and in a mouse model[46]. Regarding blockage of PD-1/PD-L1 pathways, reversal of CD8[+]T cell dysfunction and breakdown of HBV persistence were achieved in both transgenic mice and in CHB patients[47,48]. Furthermore, a PD-1/PD-L1 blockade augments the antitumor effects of peptide vaccine-induced specific CTLs in HCC patients[49]. In September 2017, the FDA has granted an accelerated approval to PD-1 checkpoint blockade (nivolumab, Opdivo) for the treatment of patients with HCC. A phase III randomized trial of nivolumab with 726 HCC patients has been launched and the estimated completion date is October 2018 (NCT02576509). However, as for only part of patients responded to PD-1/PD-L1 checkpoint therapy, looking for novel PD-1/PD-L1 interventional approaches should become a promising project in future. Here, our study confirms the therapeutic potential of miR-200c over-expression or SALL4-silencing in restoring CD8[+]T cell dysfunction (particularly, HBV-specific and memory CD8[+]T cell responses) and in reducing HBV replication, achieving almost the same therapeutic results as PD-L1 blockade treatment in our study setting. Therefore, it is possible that intrahepatic miR-200c therapy or interrupting SALL4 reactivation (SALL4 silence or SALL4 inhibitor) becomes possible alternative approaches, which deserve further testing.

Identification of reliable prognostic biomarkers that can be used to define clinical phenotypes and predict patient outcomes ensures more effective clinical treatment. Evidence indicates that serum- or tissue-specific miRNAs might act as potential diagnostic and prognostic biomarkers of liver inflammation, fibrosis, and HBV-related HCC due to their aberrant expression by hepatocytes[10,50–52]. miR-200c was recently identified as a repressor of epithelial-mesenchymal transition (EMT) via directly targeting ZEB1, while ZEB1 suppresses miR-200c transcription; thus, miR-200 and ZEB1 may oppose each other to control EMT program, tumor invasion, and metastasis[41]. Low miR-200 expression was already regarded as a predictor in disease recurrence in early stage lung cancer patients[53]. Due to its expression features and association with poor prognosis, SALL4 was also possibly a predictor of HCC. In accordance with these recent findings, this study showed a significant positive correlation of miR-200c and negative correlation of SALL4 to the survival of patients with HCC (Fig. 3, Supplementary Table 2 and 3). Patients with higher SALL4 and lower miR-200c had significantly shorter OS than patients with lower SALL4 and higher miR-200c, suggestive of possible predictive value of miR-200c and SALL4 for the survival of patients with HCC.

## Methods
**Patients and classification.** Paraffin-embedded tumor samples ($n = 98$) and peritumor samples ($n = 93$) were prospectively obtained from patients with HCC who underwent radical resection from August 2006 to September 2009 and

followed up for 4–7 years. Liver cirrhosis (n = 13) and healthy volunteers (n = 2) samples were obtained during routine hepatopathy follow-ups. All paraffin-embedded samples were collected with written and signed informed consent, which were made into tissue microarray slides (Shanghai Outdo Biotech Co., Ltd). Fresh tumor tissues were obtained from patients with HCC undergoing surgical operation at the Department of Hepatobiliary Surgery of Anhui Provincial Hospital (Hefei, Anhui, China). Supplementary Table 1 summarizes the clinical characteristics of the patients. The details of all patients have been provided according to REMARK in Supplementary Table 4. The ethical, legal, and social implications of all collected samples were approved by an ethical review board of the University of Science and Technology of China. All samples were anonymously coded in accordance with the Declaration of Helsinki. The patients' TNM stages were classified according to the seventh edition of TNM staging criteria issued by the AJCC and the UICC.

**Cell lines**. HepaRG cells were provided by Professor Fengshan Wang from Shandong University (China) cultured in a William's E medium (GIBCO/BRL) containing 15% fetal bovine serum (FBS). HepG2.2.15 cells (derived from HepG2 cells transfected with a plasmid carrying two head-to-tail copies of HBV genome DNA serotype ayw) were maintained in complete DMEM (GIBCO/BRL) supplemented with 10% FBS. HLCZ01 cells were provided by Professor Haizhen Zhu from Hunan University (China) and were cultured in collagen-coated plates, as well as with a DMEM/F12 medium (GIBCO/BRL) containing 10% FBS and ITS (Lonza). All cultures were incubated at 37 °C and 5% $CO_2$ in a humidified atmosphere. All cell lines were routinely tested against mycoplasma contamination by DAPI staining and PCR. The cell lines were not further authenticated but cultured for a limited number of passages. The mimics and inhibitors of miR-200a, miR-200c, and miR-383 were purchased from GenePharma (Shanghai, China). STAT3 inhibitor WP1006 was obtained from Selleck Chemicals (cat#S2796).

**HBV infection**. HLCZ01 or HepaRG cells were incubated with a mixture of HepG2.2.15 supernatant and fresh DMEM/F12 with a ratio of 1:1 overnight. After washing three times with PBS, HLCZ01 cells were maintained in DMEM/F12 medium, and HepaRG cells were cultured in William's E medium, and then cells were harvested at the indicated times.

**Plasmid construction**. The mouse Pd-l1 fragment was amplified from the primary hepatocytes and subcloned into XhoI and KpnI sites of the pEGFP-N1 vector. The resulting construct was confirmed by sequencing and termed pEGFP-mPd-l1. The mouse pre-miR-200c was subcloned into KpnI and XhoI sites of the pcDNA3 vector to construct the miR-200c over-expression vector. The human SALL4 fragment was amplified from HBV+ hepatoma cells and subcloned into XbaI and EcoRI sites of the pcDNA3.1 vector. The human SALL4 promoter luciferase reporter and the mouse Sall4-targeting shRNA vector were obtained from FitGene (FitGene BioTechnology CO., LTD, Guangzhou, China).

**HBV-persistent mouse model**. HBV-persistent mice were established by hydrodynamic injection of 8 μg pAAV/HBV1.2 plasmid (provided by Pei-Jer Chen, National Taiwan University College) via the tail vein into C57BL/6 male mice[21]. For delivery of plasmids into hepatocytes, 6 μg of pEGFP-mPd-l1 or pcDNA-miR-200c or Sall4 shRNA vector were hydrodynamically injected into C57BL/6 male mice. All animal experiments and protocols were approved by the Committee on the Ethics of Animal Experiments of Shandong University.

**Isolation and culture of primary mouse hepatocytes**. The C57BL/6 mice were anesthetized with sodium pentobarbital, and the liver was perfused with an EGTA solution and digested with 0.075% type II collagenase (GIBCO/BRL). The isolated hepatocytes were isolated utilizing TRIzol (Invitrogen, Carlsbad, CA, USA) according to the manufacturer's instructions. The protocol was approved by the Committee on the Ethics of Animal Experiments of Shandong University. All surgery was performed under sodium pentobarbital anesthesia, and all efforts were made to minimize suffering.

**HBV DNA analysis**. HBV viral particles in mice sera were quantified using q-RT-PCR according to the kit's instructions (Da-An, Guangzhou, China). Briefly, supernatants were collected, and mixed with DNA extraction agent (equal volume), then mixed violently for 10 s, centrifuged for 50 s and boiled for 10 min. Then, the supernatants were collected to test the copy of HBV DNA. The primers specific for the HBV S region were P1: 5′-ATCCTGCTGCTATGCCTCATCTT-3′; P2: 5′-ACAGTGGGGGAAAGCCCTACGAA-3′. The reaction was carried out for 42 cycles and performed in the iCycleriQ real-time PCR system (Bio-Rad, USA).

**ELISA**. pAAV/HBV1.2 was injected into C57BL/6 mice, and four weeks later, the sh-SALL4 or sh-CTRL was hydrodynamically injected into HBV-persistent mice. Two weeks later, the serum was harvested, and the levels HBsAg and HBeAg were detected using ELISA kits from Lan Tu Biological Systems (Quanzhou, China).

**In situ hybridization**. In situ hybridization was performed with 3′DIG- and 5′DIG-labeled probes against hsa-miR-200c (probe concentration 200 nM, 38536-15, Exiqon, Vedbaek, Denmark), 5′DIG- and 3′DIG-labeled probes against hsa-miR-200a-5p (probe concentration 500 nM, 611749-360, Exiqon, Vedbaek, Denmark) and 5′DIG- and 3′DIG-labeled probes against hsa-miR-383-5p (Probe concentration 500 nM, 610768-360, Exiqon, Vedbaek, Denmark) as previously reported. The tissue microarray sections were dewaxed in xylene and rehydrated with distilled water. This was followed by incubation with antibodies against the PD-L1 antibody (ab174838, abcam, Cambridge, USA) or the SALL4 antibody (sc-101147, Santa Cruz Biotechnology, Dallas, USA).

**Immunofluorescence and immunohistochemistry analysis**. For immunofluorescence analysis, tissues were stained with CD8 antibody (ab60076, abcam, Cambridge, USA), PD1 antibody (ab137132, abcam), SALL4 antibody (ab29112, abcam), hepatocyte specific antigen antibody (GTX73779, Genetex), CD45 antibody (MA5-17687, Thermo Scientific, Waltham,USA) and PD-L1 (ab205921, abcam) followed by Alexa Fluor 647 conjugated goat anti-rat IgG (A-21247), Alexa Fluor Plus 488 conjugated goat anti-rabbit IgG (A-32731) and Alexa Fluor Plus 555 conjugated donkey anti-mouse IgG (A-32727) (Thermo Scientific). Images were acquired on a Zeiss 710 Meta multi-photon confocal microscope (Zeiss, Oberkochen, Germany). To assess the immunostaining quantification, we analyzed the slides via an image analysis workstation (Image Pro Plus 6.0, Media Cybernetics). Mouse liver tissues were collected and embedded in OCT. Intrahepatic HBsAg, Pd-l1, or Sall4 expression was visualized by immunohistochemical staining with rabbit anti-mouse HBs Ab (Genetech, Shanghai, China) or rabbit anti-mouse Pd-l1 Ab (eBioscience, San Diego, CA, USA), or anti-Sall4 Ab (abcam, cat#57577) followed by Envision System HRP detection staining (Genetech, Shanghai, China) performed according to the manufacturer's protocol. Liver sections were stained with hematoxylin. Images were taken with an OLYMPUS microscope.

**Flow cytometry**. Surface or intracellular staining was performed utilizing the following anti-mouse mAbs or Ab controls: mouse FITC-conjugated immunoglobulin G (IgG) isotype and anti-CD8 (cat#11-0081-85, clone#53-6.7), PE-conjugated IgG isotype, anti-Pd-l1 (cat#12-5982, clone#MIH5), anti-CD69 (cat#12-0691, clone#H1.2F3), anti-CD127 (cat#12-1271, clone#ATR34), anti-IFN-γ (cat#12-7311, clone#XMG1.2), anti-CD107a (cat#12-1071, clone#1D4B), human anti-PD-L1 (cat#12-5983, clone#MIH1), anti-CD44 (cat#12-0441, clone#IM7), APC-conjugated CD62L (cat#17-0621-82, clone#MEL-14), and PE-Cy5.5-conjugated anti-CD8 (cat#35-0081-82, clone#53-6.7). All Abs were purchased from eBioscience (San Diego, CA, USA). The primary antibody was diluted in 1:100 with PBS. Dimeric H-2Kb:Ig fusion protein (BD, USA) was complexed with HBc 93-100 peptide (ANA SPEC, USA). Individual cells or mouse liver lymphocytes were stained with fluorochrome-conjugated Abs with a control IgG isotype. Intracellular staining was performed utilizing fixation and permeabilization buffers (eBioscience) according to the manufacturer's instructions. Flow cytometry was performed and data were analyzed with CellQuest software. For apoptosis detection, the mouse liver lymphocytes were stained with PE-Cy5.5-conjugated anti-CD8 mAb or control IgG isotype for 1 h. Afterward the cells were stained with Annexin V-FITC and PI for 15 min. Then, the percentage of apoptotic cells was detected by flow cytometry.

**RNA isolation and quantitative reverse transcription-PCR**. The total RNA was isolated utilizing TRIzol (Invitrogen) according to the manufacturer's instructions. The RNA concentration and quality were determined by measuring the light absorbance at 260 nm (A260) and (A260/A280) ratio, respectively. The miR-200c level was quantified by quantitative reverse transcription-PCR (q-RT-PCR) with U6 small nuclear RNA as an internal normalized reference. Bulge-Loop™ miRNA q-RT-PCR primer sets (one RT primer and a pair of qPCR primers for each set) specific for miR-200c are designed by Ribobio (Guangzhou, China). The sequences of hsa-miR-200c RT primer (ssD809230229), miR-reverse primer (ssD089261711), and 200c forward primer (ssD809230921) are proprietary.

**Western blot**. Different types of treated HLCZ01 cells were lysed in a RIPA buffer supplemented with proteinase inhibitors. 30 μg of proteins were loaded and separated on SDS-PAGE gel; then, the proteins were transferred onto a PVDF membrane, blocked in 5% (w/v) non-fat milk, and incubated with the primary antibodies. Sources of the primary antibodies were anti-STAT3 (H-190) (Santa Cruz, H-190, sc-7179), p-STAT3 (Tyr705) (Santa Cruz, sc-7993), anti-SALL1 (cat# ab130705), anti-SALL2 (Proteintech, cat#12679-1-AP), and anti-SALL4 mAb (Abcam, cat#ab61703). Protein bands were visualized by enhanced chemiluminescence (Milipore).

**Dual luciferase activity assay**. The 3′-UTR of human PD-L1 cDNA containing the putative target site for the miR-200c was inserted into the pMIR-Reporter-control vector (Promega, Madison, WI, USA) immediately downstream of the luciferase gene. HBV+HLCZ01 cells were transfected with reporter plasmids containing the 3′-UTR of PD-L1 (pMIR-Reporter-PD-L1-3′-UTR) and pRL-TK (Promega), together with 100 nM of miR-200c mimics, miR-200c inhibitors, or NCs utilizing Lipofectamine 2000 (Invitrogen) according to the manufacturer's

protocol, respectively. For the STAT3 or SALL4 target efficiency assay, the HLCZ01 cells were transiently co-transfected with a 0.1 μg pmiR-200c-Reporter working vector containing the wild-type sequence or site mutant sequence of the target gene. Luciferase activity was measured at 36 h after transfection utilizing the Dual Luciferase Reporter Assay System (Promega). Firefly luciferase activity was normalized to Renilla luciferase activity for each transfected well.

**ChIP assay.** ChIP was performed utilizing the EZ ChIP kit (Millipore) according to the manufacturer's protocol. Immunoprecipitations were carried out with the following antibodies: IgG, anti-STAT3 (abcam, cat#ab76315), and anti-SALL4 (abcam, cat#ab29112). The primer sequences for *SALL4* promoter are as follows: primer 1 forward 5′-GCCCA-GAGCAGTTATGGAAA-3′ and reverse 5′-ATTGA CACATGATGCCTGGA-3′; and primer 2 forward 5′-GATAGCTGGAGCAA GGATGG-3′ and reverse 5′-ATGAGCCCTGACAGCTGATT-3′. The primers for the miR-200c promoter are as follows: forward 5′-AGGGGTGAGACTAG GCAGGTTGG and reverse 5′-CCAGGTTGCAGTCCAAGCA.

**Statistical analysis.** Kaplan-Meier estimates of survival were used to illustrate the survival curves and to obtain the estimators of the median and survival rates for OS. Correlations between variables were evaluated using the Spearman rank correlation test for data from humans. Significant differences between groups were determined using unpaired two-tailed Student's *t*-tests unless otherwise specified; $P < 0.05$ was considered significantly different.

**Data availability.** The authors declare that the data supporting the findings of this study are available within the article and its Supplementary Information Files, or are available from the corresponding authors upon request.

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

## Acknowledgments

We wish to thank Prof. Pei-Jer Chen (National Taiwan University College) for kindly providing pAAV/HBV1.2 plasmid. We thank Prof. Haizhen Zhu (Hunan University, China) for generously providing the HLCZ01 cell line, and Dr. Olivier Govaere for the initial in situ hybridization protocol. This work was supported by the National Science Foundation of China (Nos. 81788101, 31670908, 31390433, 81771686, 91542000, 91442112, and 91442114), and Chinese Academy of Science (XDPB030301).

## Author contributions

C.S., P.L., and Q.H. designed and performed the experiments, analyzed data, and wrote the paper; M.H. and G.X. provided essential patient samples and analyzed the clinical data; J.S. and J.W. performed the experiments and analyzed the data; Z.Z., H.W., J.Z., and R.S. analyzed and interpreted the data; C.Z. and Z.T. designed the experiments, analyzed and the interpreted data, supervised the project, and wrote the paper.

## Additional information

**Competing interests:** The authors declare no competing interests.

