## [Peer Review File · Nature Communications]

Reviewers' comments:

Reviewer #1 (Remarks to the Author):

The authors of this manuscript have studied how PD-L1 expression is regulated in virus-mediated hepatocellular carcinomas using human samples. The miRNA miR-200c expression was inversely correlated with levels of PD-L1 in central tumor regions. The specificity of this miRNA mediated PD-L1 regulation was tested using two other miRNAs predicted to target PD-L1 using in silico analysis where they find specificity for miR-200c. I think this study is important and provides insights into the regulation of PD-L1.

Reviewer #3 (Remarks to the Author):

The manuscript by Sun et al reports on the detection of PD-L1 regulation through a network of complex interactions between stat3 signaling, induction of transcription factors (Sall4) and miR200c to affect PD-L1 mRNA. The author report on the correlation between these factors in HBV-associated HCC, provide evidence for the correlation between Sall4 induction after HBV infection in vitro and further provide some evidence for a role of this pathway in attenuation of virus-specific CD8 T cell responses.

The manuscripts covers aspects of immune exhaustion in HBV-associated liver cancer, regulation of PD-L1 expression in vitro and in vivo using various methodological approaches, and looks at control of HBV gene expression by CD8 T cell immunity influenced by varying levels of PDL1. The in vivo affect of Sall4 knockdown on PD-L1 expression and gain-of-function are limited.

Reviewer #4 (Remarks to the Author):

In this manuscript Cheng Sun and colleagues have investigated how PD-L1 expression is regulated in the context of virus-mediated hepatocellular carcinoma. The authors have found that miRNA miR-200c expression inversely correlated with the levels of PD-L1 expression in HCC. This is an interesting study that in part has novel aspects – however there are several control experiments needed to confirm these data. Thus, in its current form the paper needs additional work before being appropriate for publication in Nature Communications.

Major points:

Figure 1:

1. It is not clear whether the staining patterns (PDL1) found correlate with particular HCC subtypes.
2. Authors should show expression on SALL4 with co-stainings for Hepatocytes and for myeloid cells. Same holds true for PDL1. It is not clear to what extent hepatocytes and to what extent non-parenchymal cells are involved.

Figure 4:

1. The infection experiments should be repeated with HepRG (expressing NTCP for example) cells - which are a widely used in vitro model for faithful HBV replication. HLCZ01 are probably a

interesting model – but should be compared to one of the golden standards of HBV infection – also it would be confirmatory that the data can be reproduced in another model. I believe this to be particularly important.

Figure 8:

1. The CD8 and PD1 expression indicated does not really fit the pattern to be expected – it has a cytoplasmic and membranous appearance – but should be on the membrane of T-cells.

Moreover, it should be quantified – if possible – in several tissues – how often these PD1CD8+ cells directly interact with hepatocytes – are these aberrant hepatocytes?

On the whole this is an interesting manuscript – however this manuscript needs further confirmatory in vitro analyses and immunohistochemical experiments that clarify the above issues.

Point-by-point response to the referees' comments

Reviewer #1 (Remarks to the Author):

The authors of this manuscript have studied how PD-L1 expression is regulated in virus-mediated hepatocellular carcinomas using human samples. The miRNA miR-200c expression was inversely correlated with levels of PD-L1 in central tumor regions. The specificity of this miRNA mediated PD-L1 regulation was tested using two other miRNAs predicted to target PD-L1 using in silico analysis where they find specificity for miR-200c. I think this study is important and provides insights into the regulation of PD-L1.

Reviewer #3 (Remarks to the Author):

The manuscript by Sun et al reports on the detection of PD-L1 regulation through a network of complex interactions between stat3 signaling, induction of transcription factors (Sall4) and miR200c to affect PD-L1 mRNA. The author report on the correlation between these factors in HBV-associated HCC, provide evidence for the correlation between Sall4 induction after HBV infection in vitro and further provide some evidence for a role of this pathway in attenuation of virus-specific CD8 T cell responses.

The manuscripts covers aspects of immune exhaustion in HBV-associated liver cancer, regulation of PD-L1 expression in vitro and in vivo using various methodological approaches, and looks at control of HBV gene expression by CD8 T cell immunity influenced by varying levels of PDL1. The in vivo affect of Sall4 knockdown on PD-L1 expression and gain-of-function are limited.

Reviewer #4 (Remarks to the Author):

In this manuscript Cheng Sun and colleagues have investigated how PD-L1 expression is regulated in the context of virus-mediated hepatocellular carcinoma. The authors have found that miRNA miR-200c expression inversely correlated with the levels of PD-L1 expression in HCC. This is an interesting study that in part has novel aspects – however there are several control experiments needed to confirm these data. Thus, in its current form the paper needs additional work before being appropriate for publication in Nature Communications.

Major points:

Figure 1:

1. It is not clear whether the staining patterns (PDL1) found correlate with particular HCC subtypes.

Reply: Thank you for bring this important concern up. Indeed, we first found the staining patterns were highly correlated with HBsAg⁺ HCC subtype, both HBeAg⁺ and HBeAg⁻ HCC subtypes.

Further, the staining pattern have been observed in fibrolamellar, scirrhous, sarcomatoid, and lympho-epithelial-like carcinoma, suggesting there is no difference of staining patterns between all histologic HCC subtypes. However, due to limit of our sample supply, we cannot exclude the chance of HCV⁺ HCC subtype. According to your advice, we have added these in the Discussion section of our revision.

2. Authors should show expression on SALL4 with co-stainings for Hepatocytes and for myeloid cells. Same holds true for PDL1. It is not clear to what extent hepatocytes and to what extent non-parenchymal cells are involved.

Reply: Thank you for your kind advice. According to your suggestions, we have tried our best to add the co-stainings of SALL4 or PD-L1 with hepatocytes and non-parenchymal cells in our revision (Supplementary Fig. 1a and 1b). We first tried to co-stainings of SALL4 or PD-L1 with hepatocytes (hepatocyte specific antigen label) and myeloid cells (CD33 label). By using hepatocyte specific antigen antibody and CD45 antibody to label bone marrow-derived cells (non-parenchymal cells), we found that the content of PD-L1⁺ cells were higher in CT regions than those in PT regions, while the expression of PD-L1 were mainly on hepatocytes in CT regions (Supplementary Fig. 1a). SALL4 (green) expression was much higher on hepatocytes than CD45⁺ cells in CT regions, while the number of total tumor infiltrating CD45⁺ cells (magenta) in CT regions was decreased (Supplementary Fig. 1b).

On the other hand, since the different sizes of hepatocytes (cubical sides of 20-30 μm) from myeloid cells (6-8 μm), it is easier to distinguish these two group cell types on the immunohistochemistry imagings.

The images are attached as followed.

Figure legends: (a) Confocal immunofluorescence images of Hepatocyte (red), CD45 (magenta) and PD-L1 (green) staining in CT (top 2 row) and PT (bottom 2 row) regions of HCC. Original magnifications: $\times 20$, $\times 63$; scale bar: 50 μm , 10 μm , respectively.

Figure legends: (b) Confocal immunofluorescence images of Hepatocyte (red), CD45 (magenta) and SALL4 (green) staining in CT (top 2 row) and PT (bottom 2 row) regions of HCC. Original magnifications: $\times 20$, $\times 63$; scale bar: 50 μm , 10 μm , respectively.

Figure 4:

1. The infection experiments should be repeated with HepRG (expressing NTCP for example) cells - which are a widely used in vitro model for faithful HBV replication. HLCZ01 are probably an interesting model - but should be compared to one of the golden standards of HBV infection - also it would be confirmatory that the data can be reproduced in another model. I believe this to be particularly important.

Reply: Thanks for your suggestion. We repeated some experiments with in vitro HBV infection model using HepRG cell line (which expresses NTCP and is regarded as one of the golden standards of HBV infection model) to confirm the observed phenomenon and relationship among SALL4, miR-200c and PD-L1, including: Fig. 4f (about the regulation of miR-200c on PD-L1 expression); Fig. 6a, c, h (about the regulatory axis of HBV-STAT3-SALL4); Fig. 7b, f, g, h (about the negative regulation of SALL4 on miR-200c transcription) of the new version.

Figure 8:

1. The CD8 and PD1 expression indicated does not really fit the pattern to be expected - it has a cytoplasmatic and membranous appearance - but should be on the membrane of T-cells. Moreover, it should be quantified - if possible - in several tissues - how often these PD1CD8+ cells directly interact with hepatocytes - are these aberrant hepatocytes?

On the whole this is an interesting manuscript - however this manuscript needs further confirmatory in vitro analyses and immunohistochemical experiments that clarify the above issues.

Reply: Thank you for your kind advice and this point is well taken in our revision. We have revised the immunofluorescence images of SALL4, CD8 and PD-1 staining in Fig. 8g and 8h. CD8 and PD-1 expressed on the membrane of T cells, while SALL4 expressed mainly in the nucleus of hepatocytes. We found that SALL4 (green) expression was much higher in CT regions than in PT regions, while the content of tumor infiltrating CD8⁺T cells in CT regions were much lower than those in PT regions (Fig. 8g); though the number of total tumor infiltrating CD8⁺T cells (magenta) in CT regions was decreased, which is consistent with our previous findings, the content of PD-1⁺CD8⁺T cells were much higher in CT regions than those in PT regions (Fig. 8g). 3D confocal immunofluorescence analysis showed that PD-1 majorly expressed on CD8⁺T cells, and strikingly, there existed close contact between SALL4⁺hepatocytes and PD-1⁺CD8⁺T cells in CT regions (Fig. 8h).

Figure legends: (g) Confocal immunofluorescence images of SALL4 (green), CD8 (magenta) and PD-1 (red) staining in CT (top 2 row) and PT (bottom 2 row) regions of HCC. Original magnifications: $\times 20$, $\times 63$; scale bar represent $50 \mu\text{m}$ and $10 \mu\text{m}$, respectively. (i) Representative 3D confocal micrographs of CT regions of a HCC patient showing the contact between SALL4+ hepatocytes and PD-1+ CD8+ T cells. Original magnifications: $\times 63$.

According to your suggestions, we have quantified the interact rate between SALL4⁺ hepatocytes and PD1+CD8⁺ cells in 6 HCC patients. Five random $20\times$ fields of view were selected from each HCC patient and contact rate was scored manually in a blinded manner. The details of results were shown below.

Reviewers' comments:

Reviewer #3 (Remarks to the Author):

The manuscript has been improved by the additional experiments performed by the authors. The experiments with hepatocytes demonstrate a clear regulation of PDL1, yet definitive experimental proof for upregulation of PDL1 in HBV-infected cells is still lacking. It also remains unclear why only some cells in the immunohistochemistry show a positive signal for PDL1 whereas actually most of the hepatocytes under in vitro conditions do.

Reviewer #5 (Remarks to the Author):

Cheng Sun et al studied regulation of PD-L1 expression in HCC patients with HBV infection, showing that miR-200c directly inhibits HBV-mediated PD-L1 expression. It has been reported that miR-200c is inhibited by SALL4. Therefore, SALL4-induced expression in HBV context leads to increased PD-L1 expression. These results are novel and highlighting a new mechanism of HBV-related immune escape through PD-L1-induced expression. The authors added substantial new data and reproduced their observation in another cellular model.

The mechanisms of PD-L1 induced expression are very interesting and novel. However, several points need to be more documented to emphasize their study for publication in Nature Communications.

Major points:

1) PD-L1 expression by tumor cells was reported in several studies to reach 17 to 25% of analyzed HCC. The threshold of PD-L1 expression is usually $\geq 1\%$ for tumor cells. PD-L1 is also expressed by inflammatory cells as macrophages in HCC. PD-L1 expression by immune cells has been described to be found in 75% of HCC. The reported data does not allow to distinguish cell types that express PD-L1. Indeed, only scanning the slides after PD-L1 and SALL4 staining is not sufficient to bring such information in the cohort studied.

2) In the same line, it is unclear on Liver Cirrhosis compared to healthy subjects whether PD-L1 was directly expressed by hepatocytes and/or immune cells infiltrating a liver with chronic inflammation (Supplementary figure 3). Additional information may be important to distinguish the direct role of HBV on hepatocytes and indirect role on the immune microenvironment.

3) Furthermore, data with clinical features from patients (both for HCC and Liver cirrhotic patients) should be presented (e.g Child-pugh score, HBV status regarding HBs, HBc, HBe antigens, HBV DNA PCR, and Fibrosis / Activity scores). These data clinical parameters will be very helpful to support the relevance of the murine model used.

Minor points

1) In the result part: Other studies showed a relationship between PD-L1 expression by tumor cells and aggressiveness parameters including the tumor differentiation grade and/or vascular invasion. It would be of high interest to have these data regarding the prognostic impact of SALL4, miR200c and PD-L1 expression in univariate Cox regression (supplementary table 2). A multivariate Cox regression analysis with other classical prognosis factors would confirm the prognostic impact of these markers.

2) In discussion part, the followed sentences are unclear: "We found that the staining patterns of PD-L1 were highly correlated with HBsAg+ HCC subtypes [...] between all histologic subtypes". To my knowledge, in the present manuscript, HBV antigens are only measured in HBV persistent mice without HCC, and pathological HCC subtypes are not indicated in the results or in the supplementary figures.

Response to Reviewers' comments:

Reviewer #3 (Remarks to the Author):

The manuscript has been improved by the additional experiments performed by the authors. The experiments with hepatocytes demonstrate a clear regulation of PDL1, yet definitive experimental proof for upregulation of PDL1 in HBV-infected cells is still lacking. It also remains unclear why only some cells in the immunohistochemistry show a positive signal for PDL1 whereas actually most of the hepatocytes under *in vitro* conditions do.

Reply: Thank you for your kind comments. Although not all the hepatocytes express PD-L1 in HBV⁺HCC patients, as shown in Fig. 1b (immunohistochemistry staining), it is clearly that most hepatocytes express high levels of PD-L1 in center tissues, whereas PD-L1 usually expresses on myeloid cells but not most hepatocytes in peritumor tissues of HBV⁺HCC patients. In consistent with this, previous studies by other labs also showed that PD-L1 expression is approximately 17-25% on tumor cells in HCC patients (Hepatology. 2016; 64 (6): 2038-2046, Clin Cancer Res. 2009;15(3):971-979). We think multiple factors, such as the abnormal levels of IL-4, IL-27, TGF- β , TNF- α and IFN- γ in the microenvironment (particularly in centretumor) of HBV-infected HCC, may lead to the different expression of hepatocyte PD-L1. So, HBV is possibly not the only factor to regulate PD-L1 expression that is why not all hepatocytes express PD-L1 in tumor microenvironment. In addition, as shown in Fig. 4 and Fig. S4, we observed that not all HBV⁺hepatocytes express PD-L1 *in vitro*.

Reviewer #5 (Remarks to the Author):

1. Figure 1:

It is not clear whether the staining patterns (PDL1) found correlate with particular HCC subtypes.

Please add reference or data figure related to HCC subtypes including results obtained on HBsAg+ HCC. Indeed, this point is mentioned in p8 line 317 but without support.

Reply : Thank you for your kind advice. According to your suggestion, we have added data related to HCC subtypes in our Supplementary Table 4, including HBV, HCV, HBsAg, HBsAb, HBeAg, HBeAb and HBcAb as well as different histologic subtypes. As you can see, 100% of our HCC cohort was HBsAg⁺ patients. For HBV-related HCC subtypes, we found that PD-L1 were highly correlated with HBV⁺ HCC subtype but not HCV, which is consistent with our observation in HBV⁺hepatocytes of HBV-persistent mice (Supplementary Fig. 3) and the previous report from Xie Z who showed the correlation of *in situ* PD-L1 expression to HBV load in patients with chronic HBV infection (Immunol Invest. 2009;38(7):624-638). There has no difference been observed between patients with different HBeAg expression and HBeAb serum level. For histology-related HCC subtypes, the PD-L1-positive staining has been found in most types of HCC, including trabecular, pseudoglandular, fibrolamellar, scirrhous and sarcomatous carcinoma, regardless of histologic subtypes. We have also stratified the patients according to the UICC-TNM classification. As shown in the figure below, similar staining pattern was observed in HCC patients with different morphological subtypes; moreover, we observed that late-stage HCC patients (TNM IV, pathology grading 3) maintained higher levels of PD-L1 compared with those of early-stage HCC patients (TNM I, pathology grading 1). We added these in the Discussion section of our revision.

2. Figure 4:

The infection experiments should be repeated with HepRG (expressing NTCP for example) cells - which are a widely used in vitro model for faithful HBV replication. HLCZ01 are probably an interesting model – but should be compared to one of the golden standards of HBV infection – also it would be confirmatory that the data can be reproduced in another model. I believe this to be particularly important.

The authors provided convincing data to answer the question.

Reply : Thanks.

3. Figure 8:

The CD8 and PD1 expression indicated does not really fit the pattern to be expected – it has a cytoplasmatic and membranous appearance – but should be on the membrane of T-cells. Moreover, it should be quantified – if possible – in several tissues – how often these PD1CD8+ cells directly interact with hepatocytes – are these aberrant hepatocytes? On the whole this is an interesting manuscript – however this manuscript needs further confirmatory in vitro analyses and immunohistochemical experiments that clarify the above issues.

Confocal microscopy shows indeed a cytoplasmic staining of CD8 and PD1 which is not expected, probably due to non-specific immunostaining.

Reply : Thank you for the kind suggestion. Since Fig. 8h is a three dimensional photograph (not same as a membrane “ring” in 2D), beautifully showing that four PD-1⁺ CD8⁺ T cells are contacting with one SALL4⁺ hepatocytes in a global space, the membrane staining of CD8 and PD-1 could not be exhibited as a “ring” and not distinguish from cytoplasmic staining. Also, the 2D photographs in Fig. 8g clearly showed the membrane staining of CD8 and PD-1 of CD8⁺T cells. So if you put Fig. 8g (2D imaging) and 8h (3D imaging) together, you can clarify the interaction between SALL4⁺Hepatocyte and PD-1⁺T cells. Further, all confocal photographs in our study were captured with consistent parameters, and negative and positive controls have been performed to exclude nonspecific staining in each experiment.

Taking into account possible ambiguities 3D image may cause, we have strip the Fig. 8h (3D data) out of the revised version as suggested by editor. Or it may take the reviewer’s advice to determine whether it is kept or stripped out.

4. Authors should show expression on SALL4 with co-stainings for Hepatocytes and for myeloid cells. Same holds true for PDL1. It is not clear to what extent hepatocytes and to what extent non-parenchymal cells are involved.

Requested experiments were performed. However, the presented data are not convincing. Firstly, in panel a PDL1 seems to be expressed only on one hepatocyte “in center tumor region”. The zoomed region shows a PDL1 staining while no staining was observed in the corresponding selected area at a lower magnification. Secondly, in the panel b, no CD45 staining was seen in the selected center regions while this is obvious in the selected corresponding region in panel a.

Reply : Thank you for your comments. For panel a, as you can see, there are more than one PD-L1-expressing hepatocytes in the upper photograph (the figure below showed the screenshot of panel a), although it seems only one PD-L1-staining hepatocyte in the selected area (white box) with magnification. This might indicate a fact that not all the hepatocytes express PD-L1 in HCC.

For panel b, the selected region is not the corresponding region in panel a, which can be shown in the brightfield photos in each panel. Indeed, we found that the numbers of infiltrating CD45⁺ cells in the center region were significantly decreased than those in the peritumor regions of HCC patients. This can be observed in both panel a and panel b in Supplementary Fig.1. This phenomenon is consistent with other reports (e.g. J Hepatol. 2017 Apr;66(4):743-753). So, our results suggest that PD-L1 mainly expressed on hepatocytes but rarely myeloid cells in CT region, while it mainly expressed on myeloid cells in peritumor regions of HCC, if you see immunohistochemical figures (Fig. 1, Fig. 2, and Fig. S2) with attention to the morphology and the cell size specialized for hepatocytes.

Indeed, we agree that our image in Fig. S1 may not clearly display the PD-L1 patterns on hepatocytes or myeloid cells. We have strip the Fig. S1 out of the revised version as suggested by editor. And we added some discussion about this issue in section of Discussion of the revised manuscript.

Reviewer #5 (Remarks to the Author):

Cheng Sun et al studied regulation of PD-L1 expression in HCC patients with HBV infection, showing that miR-200c directly inhibits HBV-mediated PD-L1 expression. It has been reported that miR-200c is inhibited by SALL4. Therefore, SALL4-induced expression in HBV context leads to increased PD-L1

expression. These results are novel and highlighting a new mechanism of HBV-related immune escape through PD-L1-induced expression. The authors added substantial new data and reproduced their observation in another cellular model.

The mechanisms of PD-L1 induced expression are very interesting and novel. However, several points need to be more documented to emphasize their study for publication in Nature Communications.

Major points:

1) PD-L1 expression by tumor cells was reported in several studies to reach 17 to 25% of analyzed HCC. The threshold of PD-L1 expression is usually $\geq 1\%$ for tumor cells. PD-L1 is also expressed by inflammatory cells as macrophages in HCC. PD-L1 expression by immune cells has been described to be found in 75% of HCC. The reported data does not allow to distinguish cell types that express PD-L1. Indeed, only scanning the slides after PD-L1 and SALL4 staining is not sufficient to bring such information in the cohort studied.

Reply: Thank you for your comments. We have observed the express pattern of PD-L1 in HBV⁺HCC patients. As shown in Fig. 1, Fig. 2, and Fig. S2, PD-L1 mainly expressed on hepatocytes but rarely expressed on bone-marrow-derived cells (such as Kupffer cells, macrophages or immune cells) in CT region, while it mainly expressed on bone-marrow-derived cells in peritumor regions of HCC and in liver cirrhosis, with attention to the morphology and the cell size specialized for hepatocytes.

2) In the same line, it is unclear on Liver Cirrhosis compared to healthy subjects whether PD-L1 was directly expressed by hepatocytes and/or immune cells infiltrating a liver with chronic inflammation (Supplementary figure 3). Additional information may be important to distinguish the direct role of HBV on hepatocytes and indirect role on the immune microenvironment.

Reply: Thank you for your suggestions. We have added higher magnification of photos in our revised Supplementary Figure 2, from which you may easily distinguished that PD-L1 expressions in liver cirrhosis and healthy subjects were mainly on bone marrow-derived cells, with attention to the morphology and the cell size specialized for hepatocytes. Regarding indirect role on the immune microenvironment, we answer this in “Reply to Reviewer #3” as followed.

“Although not all the hepatocytes express PD-L1 in HBV⁺HCC patients, as shown in Fig. 1b (immunohistochemistry staining), it is clearly that most hepatocytes express high levels of PD-L1 in center tissues, whereas PD-L1 usually expresses on myeloid cells but not most hepatocytes in peritumor tissues of HBV⁺HCC patients. In consistent with this, previous studies by other labs also showed that PD-L1 expression is approximately 17-25% on tumor cells in HCC patients (Hepatology. 2016; 64 (6): 2038-2046, Clin Cancer Res. 2009;15(3):971-979). We think multiple factors, such as the abnormal levels of IL-4, IL-27, TGF- β , TNF- α and IFN- γ in the microenvironment (particularly in centretumor) of HBV-infected HCC, may lead to the different expression of hepatocyte PD-L1. So, HBV is possibly not the only factor to regulate PD-L1 expression that is why not all hepatocytes express PD-L1 in tumor microenvironment.”

3) Furthermore, data with clinical features from patients (both for HCC and Liver cirrhotic patients) should be presented (e.g Child-pugh score, HBV status regarding HBs, HBc, HBe antigens, HBV DNA PCR, and Fibrosis / Activity scores). These data clinical parameters will be very helpful to support the relevance of the murine model used.

Reply: Thank you very much for bring this important concern up. We have added all clinical parameters in details of every patient in Supplementary Table 4.

Minor points

1) In the result part: Other studies showed a relationship between PD-L1 expression by tumor cells and aggressiveness parameters including the tumor differentiation grade and/or vascular invasion. It would be of high interest to have these data regarding the prognostic impact of SALL4, miR200c and PD-L1 expression in univariate Cox regression (supplementary table 2). A multivariate Cox regression analysis with other classical prognosis factors would confirm the prognostic impact of these markers.

Reply: Thanks to your suggestions. We have added the prognostic impact of these factors with univariate and multivariate cox regression analysis in section of Results and Supplementary Table 2 and 3 in our revised version.

2) In discussion part, the followed sentences are unclear: “We found that the staining patterns of PD-L1 were highly correlated with HBsAg+ HCC subtypes [...] between all histologic subtypes”. To my knowledge, in the present manuscript, HBV antigens are only measured in HBV persistent mice without HCC, and pathological HCC subtypes are not indicated in the results or in the supplementary figures.

Reply: Thank you for your kind advice. According to your suggestion, we have added new data related to HCC subtypes in our Supplementary Table 4, including HBV, HCV, HBsAg, HBsAb, HBeAg, HBeAb and HBcAb as well as different histologic subtypes. As you can see, 100% of our HCC cohort was HBsAg⁺ patients. For HBV-related HCC subtypes, we found that PD-L1 were highly correlated with HBV⁺ HCC subtype but not HCV, which is consistent with our observation in HBV⁺ hepatocytes of HBV-persistent mice (Supplementary Fig. 3) and the previous report from Xie Z who showed the correlation of *in situ* PD-L1 expression to HBV load in patients with chronic HBV infection (Immunol Invest. 2009;38(7):624-638). There has no difference been observed between patients with different HBeAg expression and HBeAb serum level. For histology-related HCC subtypes, the PD-L1-positive staining has been found in most types of HCC, including trabecular, pseudoglandular, fibrolamellar, scirrhus and sarcomatous carcinoma, regardless of histologic subtypes. We have also stratified the patients according to the UICC-TNM classification. As shown in the figure below, similar staining pattern was observed in HCC patients with different morphological subtypes; moreover, we observed that late-stage HCC patients (TNM IV, pathology grading 3) maintained higher levels of PD-L1 compared with those of early-stage HCC patients (TNM I, pathology grading 1). We added these in the Discussion section of our revision.

REVIEWERS' COMMENTS:

Reviewer #5 (Remarks to the Author):

The authors satisfactorily addressed all questions.

The added experiments significantly improved the manuscript.

Response to Reviewers' comments:

Reviewer #5 (Remarks to the Author):

The authors satisfactorily addressed all questions.

The added experiments significantly improved the manuscript.

Reply: Thanks a lot.